# Structural patterns for transmedia storytelling

**Ryan Javanshir** \*, **Beth Carroll, David Millard**

University of Southampton, Southampton, England, United Kingdom

\* ryanjavanshir@hotmail.co.uk

## Abstract

Transmedia storytelling involves telling a story using multiple distinct media. The remit of stories that fall under this broad definition is vast, consequently causing theorists to examine different phenomena using tools that are not suitable for all forms of transmedia storytelling. The lack of critical tools means we are unable to describe, compare and analyse different experiences using common language. In this paper, we present our model that can be used to identify the fundamental structural features of a variety of transmedia storytelling forms. We illustrate its usage with twenty case studies and discuss how three groups of patterns emerge which can be identifiable in all transmedia stories. These patterns can be used to extend transmedia language and help form taxonomies, by identifying common patterns and their usages amongst various forms of transmedia stories.

**Data Availability Statement:** All relevant data are within the manuscript and its Supporting Information files.

**Funding:** EPSRC Funded.

## 1 Introduction

There are many ways we can experience a fictional world, its events, characters and stories in the present day. We not only consume stories from mono-media such as watching films, playing games and reading books, but we also experience them by traversing multiple media, known as transmedia storytelling. In Convergence Culture, Henry Jenkins describes transmedia storytelling as a "process where integral elements of a fiction get dispersed systematically across multiple delivery channels for the purpose of creating a unified and coordinated entertainment experience. Ideally, each medium makes its own unique contribution to the unfolding story"[1].

The complexity that comes with telling stories this way brings with it increased difficulty in understanding the structure of these experiences. We can study films by watching them, perhaps commenting on the mis en scene, their portrayal of characters by their dialogue and the way they act, how scenes are structured and transition from one to the next, or the angle of the camera. We can study games by playing them, considering how the player's interaction affects the fictional world, the user interface and its role in communicating information to the player, or how the levels and environments are designed. But how do we study a story that is dispersed on multiple media, told using websites that evolve over time, social media and forums that allow role-play, video sharing websites, PDF documents, image websites, games, films, books, comics and television? How do we, in Long's words, "close read"[2] such experiences? What is considered to be a transmedia story in the first place?

In a blog post by Andrea Philips, she comments that "that indie art scene that started with alternate reality games is, well, it's over"[3], expanding that the concept of transmedia

**Competing interests:** The authors have declared that no competing interests exist.

storytelling is clearly not dead, but has changed. The buzzword that gained prominence in 2006, after it was coined by Henry Jenkins, has become an umbrella term used to describe wildly diverse forms such as; escape rooms, mixed reality games, secret cinema, virtual reality experiences, second screen apps, digital exhibitions and complex franchises to name a few[4]. This idea is reinforced by recent transmedia literature that considers attractions, television, photography, sports, journalism, games and music to be within its remit[5]. The problem here is that we have defined transmedia storytelling so broadly that it sometimes becomes unfair to justify comparisons between vastly different experiences. How can we compare an escape room to a second screen app when both experiences use different media, progress the story differently and ask for different requirements from their audiences?

## 1.1 Aims and contributions

Our first contribution involves using our model[6] to conduct a structural analysis of twenty transmedia stories. Our second contribution encompasses these analyses by identifying metrics that can be associated with what we call patterns, or structural features of transmedia stories. Our third contribution explores how different forms of transmedia storytelling utilise these patterns to achieve certain effects and how well these patterns map to the objectives of a story.

## 2 Background

In this section, we consider some of the definitions of both transmedia and media before moving on to related work in transmedia modelling and methods for transmedia categorisation.

For the past decade, objects of study that overlap around transmedia have emerged and contributed to the ongoing debate surrounding how these experiences can be defined, and what tools are most appropriate for critically analysing them [7], [8], [9] [2], [10]. In their efforts to delimit transmedia stories from other experiences, a multitude of definitions have been made; transmedia fictions[11], supersystems of transmedia intertexts[12], digitexts[13], and multimedia strategies[14].

Wide definitions such as that proposed by Lisbeth Klastrup and Susana Tosca use the term "transmedial worlds" to describe abstract content systems that use a variety of media forms, where the "audience and designers share a mental image of the "worldness" (a number of distinguishing features of its universe)"[15]. Elizabeth Evans applies theory from television studies when considers transmedia to use the concept of flow, "a collection of different segments of content that are brought together into a whole larger than any individual segment and guided by an ever present, though potentially invisible, time-based organisational structure" [5]. Evans identifies flow's contribution in highlighting "the blurred boundaries that exist between different kinds of television content and how audiences need to navigate their way through those blurred boundaries" [5]. This brings us to the next question: what is a medium and how can you make the boundaries clearer?

Media theorist Dan Laughey defines media as a "means of communication"[16]. Though these means can also have within them additional means of communication as Marshall McLuhan argues, "the content of any medium is always another medium"[10]. Pratten employs a four-layered definition of media. A channel describes the basic sensory communication methods such as audio, media is the embodiment of a channel such as a file, platforms support media such as YouTube, and devices allow the audience to access the platforms such as a smartphone[17]. However, this does not include media that are technically the same but culturally different e.g. comics use the same channels as novels. In furthering his definition of transmedia, Jenkins comments that each medium of a transmedia story should make "a

distinctive and valuable contribution to the whole"[1]. This technical approach taken with Jenkins' subjective approach will afford us an extra tool when determining the boundary between one medium and another.

Given these definitions, we can determine a medium by its unique means of communication either technically or culturally, and whether or not it makes a distinct and valuable contribution to the transmedia story as a whole. Subsequently, we consider any story that is told through more than one medium as a transmedia story.

## 2.1 Related work

The language used to describe transmedia stories, and consequently the taxonomies associated with transmedia, varies across disciplines [18]. As a result, theories have emerged that attempt to categorize transmedia stories in various ways, through their structural, cultural or thematic categories. Espen Aarseth uses synchronicity as a parameter with which to categorise transmedia experiences depending on when they are published relative to each other, including synchronous experiences that release content at similar times, and asynchronous experiences that release content sequentially.[19] Robert Pratten identifies three types of transmedia story; franchise, portmanteau and complex, the describe the relationship between the media. The first describes a storyworld that is conveyed in multiple stories using multiple media, the second describes a single story that consists of multiple media and the third is a combination of these two [17]. Similarly, Andrea Phillips classifies stories depending on how linked the media are to one another, and their narrative dependency[20]. Focusing on franchises, Jai E. Jung describes a transmedia franchise taxonomy that focuses on individual texts and their narrative-based temporal relation to other texts in the franchise e.g. prequels, sequels, interquels, midquels etc.[21].

In addition to proposed language, models have been developed that seek to identify features of transmedia stories, which in turn can be used to distinguish between various forms. Firstly, a model developed by Mariana Ciancia documents which channels were used, the overall story as a written account, and an action flow showing how events are conveyed to participants.[22] Similarly, Renira Rampazzo Gambarato has developed an analytical model that identifies a number of features such as narrative, characters, and structure and context which provides a platform for practicable questions to be asked that reveal details about these features. They argue that their model "depict, in a simpler or deeper manner, the essential structure of transmedia projects" [5]. "The Variable Model" produced by Espen Aarseth, looks at features of narrative games such as the world, its objects, agents and events. The model includes a scale that goes from one end, the narrative pole to the other, the ludic pole, enabling analysts to see whether any given game tends to be more narratively or gameplay driven in any given feature. [23] A more visual approach taken by Marc Ruppel applies graph theory to transmedia "networks", with media occupying the role of vertices, and the relationships between them fulfilling the role of network connections or as he calls them, migratory cues. Ruppel argues that this mapping allows a close reading of transmedia stories to occur by allowing for characteristics to be identified, such as the amount of links to and from a medium. These characteristics can then lead to the identification of specific network motifs and can be used as a basis for a taxonomy of transmedia networks[24].

## 3 Methodology

Previous approaches offer insight into what characteristics can be used to form a transmedia taxonomy. However, instead of focusing on a particular form of transmedia such as franchises or single story transmedia experiences, we intend to extend this theory by using our model as

**Table 1. Instance state and interactivity.**

| Instance Characteristics | Passive | Active |
|---|---|---|
| **Live** | e.g. live theatre | e.g. live action role play |
| **Static** | e.g. YouTube video | e.g. video game |

a vehicle to illustrate various characteristics identifiable in all forms of transmedia storytelling, using twenty case studies. In this section, we briefly describe our model, and how the case studies were selected using our selection criteria.

## 3.1 The model

In [6], we described the model and applied it to two ARGs to illustrate its usage. Our model is based around the concept of a *channel*, a subset of a media channel that is defined by its boundary e.g a website W1. *Channels* have *instances* to illustrate a changed state e.g. a website update W1(1), with these instances having *link*s, *interactivity* and *state* associated with them. Links occur between instances, such as from a website to a film. Interactivity is either *passive*, if the audience passively consume content e.g. a film and *active* when they take on a role in the story e.g. a game where you play as a character. A state can be either *live* if it cannot be accessed in its original form after it has occurred e.g. a play and *static* if it can e.g. a film. Instances are associated with *scenes* that signify the narrating time. If an instance is part of that scene, it means that during that time, only those instances can be accessed. Table 1 shows a summary of instance characteristics.

The model therefore models a transmedia story as a set of channels, with each channel comprised of a sequence of instances that are part of a scene. Each instance contains links, interactivity/state values and an optional description. This model can be used in both written (shown in Table 2), and visual form, with the visual form being used to illustrate the model in this paper.

Fig 1 is a visual form example of a transmedia story that begins with website W1, which has been updated 4 times. In the first update (instance), it links to the second website which in turn links to a game. The game then links to a live stream where the story ends.

Since [6], we have updated the model with two additional parameters to accommodate our expanded set of case studies. Below are the updates that have been utilised in this article.

- Non-italic channel name means the channel is standalone. A standalone channel is one that can be experienced on its own as a story, regardless of all other channels in the experience.

- Bold italic channel name means the channel is a subsidiary channel. A subsidiary channel is one that relies on another channel either technologically e.g. audio sync technology that relies on a film, or via a link e.g. a website with an obscure URL that relies on a hyperlink from another known website.

## 3.2 How the case studies were selected

Twenty case studies (shown in Table 3) were selected. Although there are many different forms of transmedia storytelling, we opted to select case studies that fell under one of six

**Table 2. Model of a single channel.**

| Name of Channel (i.e. W1) | | | | | |
|---|---|---|---|---|---|
| Instance | Scene | Links | Interactivity | State | Description |

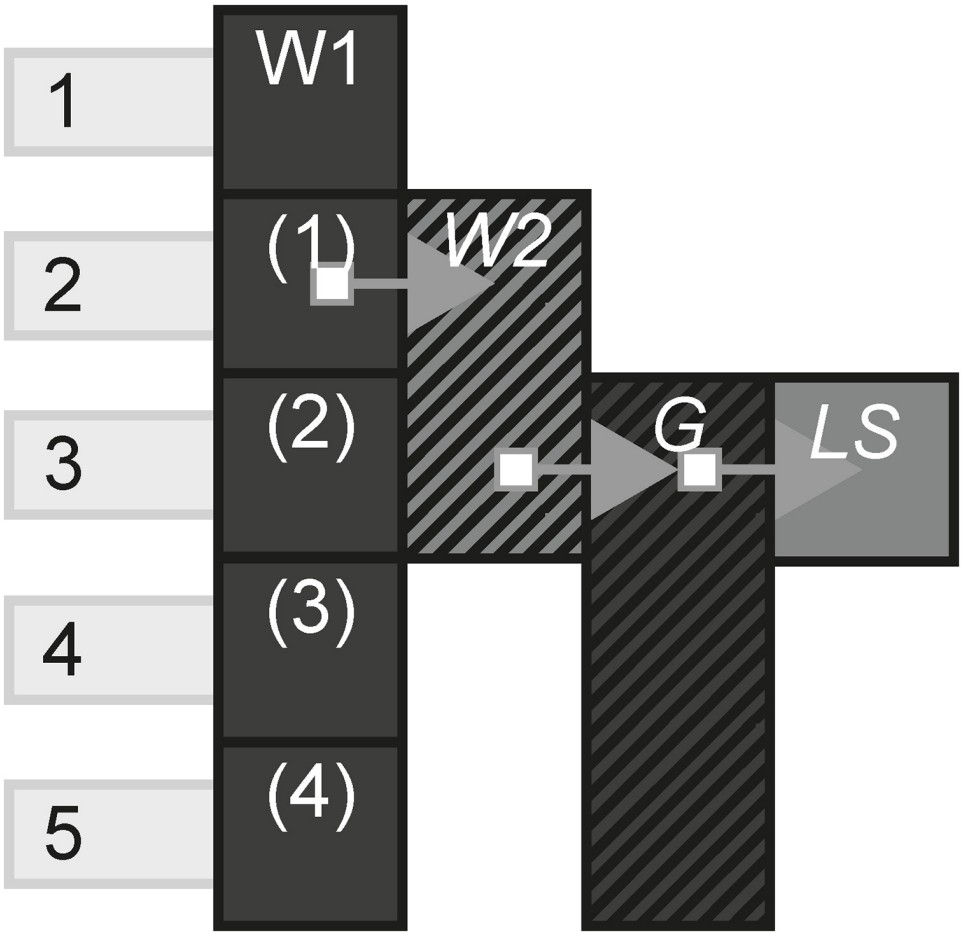

**Fig 1. A transmedia story example.**

forms; interactive films/second screen apps, Alternate Reality Games (ARGs), media franchises, escape rooms, table-top RPGs and exhibits. These forms were identified using the sources above as well as how producers saw their story and third party bodies included in our additional sources material. Though this list of forms is not exhaustive, we felt that by selecting examples that fell under this remit, our potential list of case studies would succeed in offering the desired level of diversity required for this study. We then filtered this list again using a selection criterion that focused on practicality and diversity.

**Practicality.** The ephemeral nature of many transmedia stories meant that data such as third party post mortems, blogs, videos, wikis, summaries and any other archived content would be used where that experience could not be consumed first hand. Every effort has been taken to minimise errors and subjective distortion by taking data from a variety of sources by different authors and domains.

**Channel diversity.** We wanted to test the scope of the model by applying a wide range experiences to it, finding out whether the model is capable of distinguishing between different forms of transmedia storytelling. Transmedia stories use a variety of techniques such as a range of media combinations, differing levels of audience interaction and different relationships between media. We attempted to select case studies that were different to each other in these ways so that distinctions could be made.

**Table 3. Selected case studies.**

| Title | Author | Form |
|---|---|---|
| 19 Reinos | HBO | ARG |
| APP | 2CFilm | Second screen film |
| Bandersnatch | Netflix | Interactive film |
| *Change The Record* | Xciting Escapes | Escape room |
| *Defenders of the Triforce* | Nintendo | Escape room |
| Dexter | CBS Television | ARG |
| *Dungeons and Dragons* | Wizards Coast | Table top role-play |
| Game of Thrones | HBO | Media franchise |
| Harry Potter | Bloomsbury, Warner Bros. Pictures | Media franchise |
| Her Story | Sam Barlow | Videogame |
| Overwatch | Blizzard | Videogame |
| Pirates of the Caribbean | Walt Disney Pictures | Media franchise |
| Pokémon | Nintendo | Media franchise |
| Prometheus Second Screen | 20$^{th}$ Century Fox | Second screen film |
| Prometheus Campaign | 20$^{th}$ Century Fox | ARG |
| *Roman Baths* | Bath & North East Council | Exhibition |
| The Black Watchmen | Alice & Smith | Videogame |
| The Matrix | Warner Bros. Pictures | Media franchise |
| Westworld Campaign | Warner Bros. Television | ARG |
| Why So Serious? | Warner Bros. Pictures | ARG |

However, we also took an opportunistic approach and deviated from our list, extending it with local live events with which we could get first-hand experience (in italics below). We felt that this approach offered a wider variety of stories to be included into our list of case studies. Although it is recognised that for practicality we had a limited choice with such experiences.

## 4 Case studies

This section includes twenty cases that we selected using the selection criteria. Each subsection includes a brief description of the experience, followed by its model.

### 4.1 19 Reinos

19 Reinos (19 Realms in English) was an ARG with heavy role-playing elements based on the popular HBO series Game of Thrones (GoT) which in turn was based on a book A Song of Ice and Fire by George RR Martin. It ran for 10 weeks in 2004, at the same time season 4 of GoT was released. Media such as TV, a web series, social media, websites and live events were utilised for the telling of the story. Fig 2 shows the model of 19 Reinos.

### 4.2 APP second screen

APP is a Dutch film about a young woman who, after a party, finds an unknown app has downloaded on to her phone. Over time the app starts to terrorise the woman in increasingly terrifying ways. On its release, the producers encouraged the audience to bring their phone to the theatre, download the app and have it sync to the movie. The app then revealed various media content to the audience as the film was playing, that linked to the current scene on screen. Fig 3 shows the model of APP.

# Game of Thrones: 19 Reinos

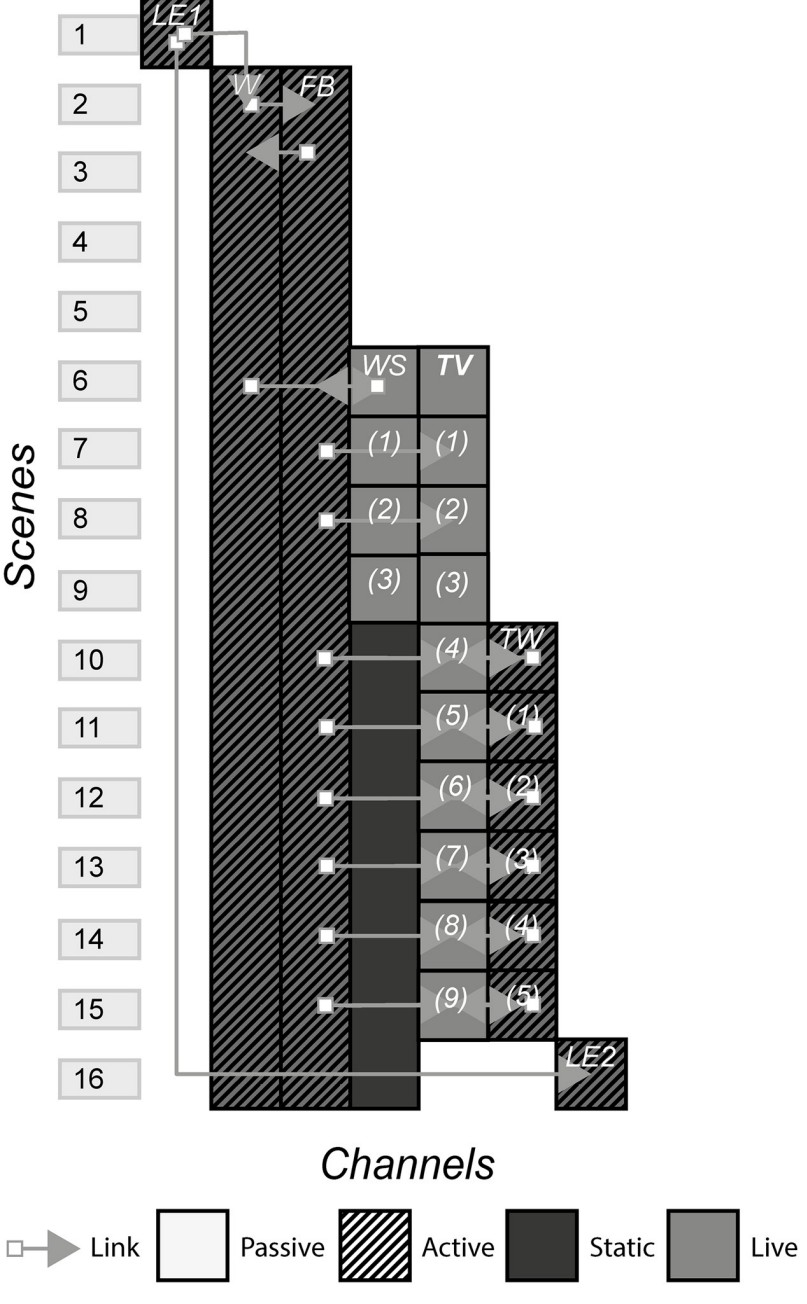

**Fig 2. GoT: 19 Reinos.**

## 4.3 Change The Record

Change The Record is an 'escape room' where a small group of people get briefed on a thematic objective before getting locked in a room. Once in the room, the group has to solve puzzles to ultimately complete the mission, find the key and 'unlock' the door. The theme for this experience involves the audience role-playing as British intelligence operatives in 1989 that

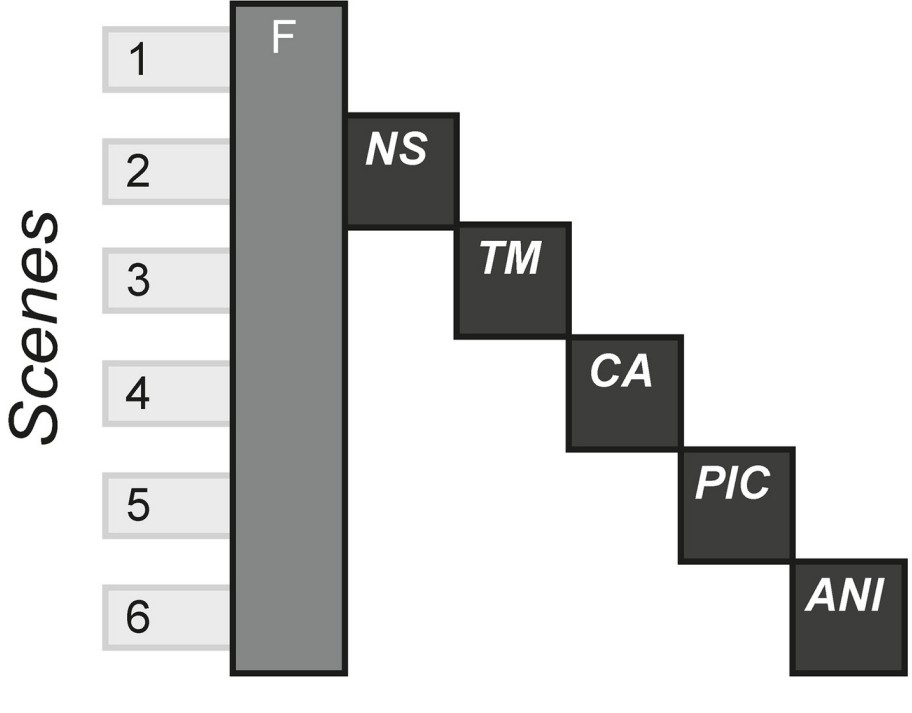

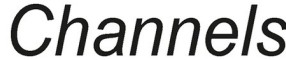

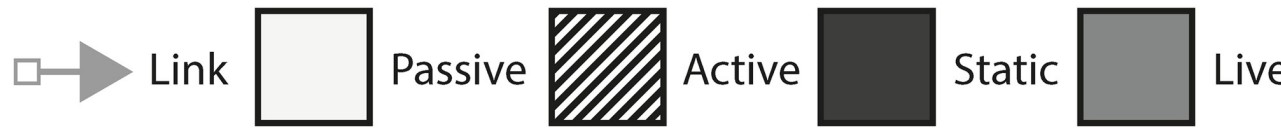

**Fig 3. APP.**

have been tasked with uncovering the Communist enemy agents' plans by searching their undercover record shop, located in Southampton. Fig 4 shows the model of Change The Record.

## 4.4 Defenders of the Triforce

In 2017, Nintendo sold tickets for a limited run of their themed immersive live experience, Defenders of the Triforce. The experience was themed on The Legend of Zelda, a video game series spanning many generations of Nintendo consoles. The game series involves questing and puzzle solving in a fantasy world, with the aim of destroying the ancient evil known as

# Escape Room (Change The Record)

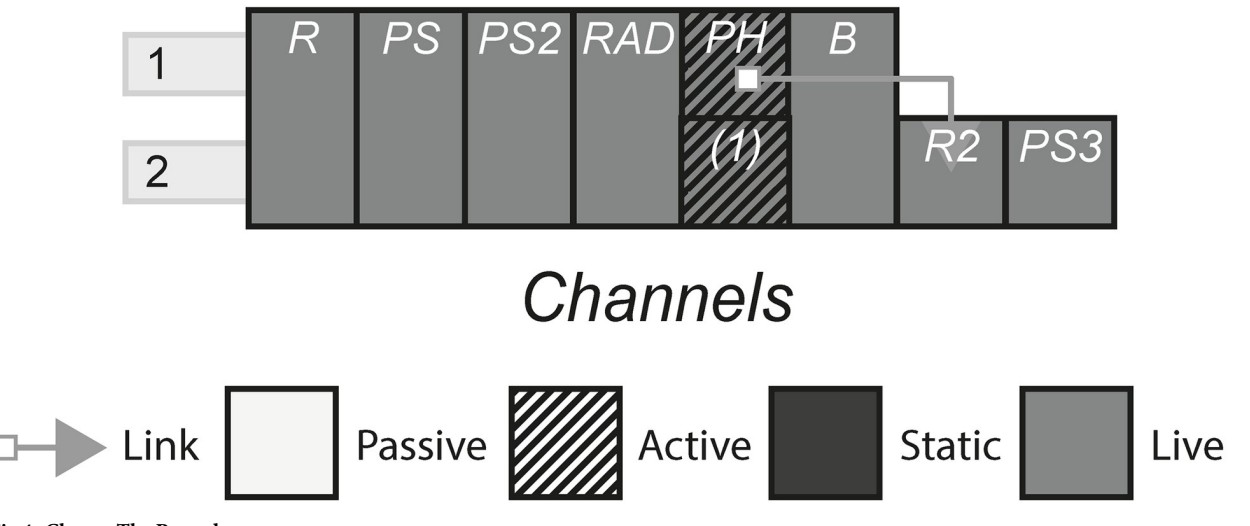

**Fig 4. Change The Record.**

Ganon. The live experience invited players to take the role of the heroes who must defeat Ganon by completing several objectives such as word puzzles, physical challenges and interaction with the live actors who were playing as characters. Fig 5 shows the model of Defenders of the Triforce.

## 4.5 Dexter ARG

Made to promote the upcoming Dexter television series, the Dexter ARG involved the audience attempting to hunt down a murderer from the clues they had been given by a detective. In the end, the audience as a collective had to decide who ultimately 'won', the detective or the murderer. Fig 6 shows the model of Dexter ARG.

## 4.6 Table top role-playing game (e.g. Dungeons and Dragons)

Tabletop role-playing games, popularized by Dungeons and Dragons that was first published in 1974, are games that involve a 'dungeon master' who interprets a manual of lore and role-playing rules to create a hand crafted narrative for a party of players. Fig 7 shows the model of an example table top role-playing game.

## 4.7 Game of Thrones franchise

First published as a series of books under the name A Song of Ice and Fire, Game of Thrones is a television show that has reached global fame, arguably like no other television show has by creating a deep impact into popular culture. Mostly based off of the same content as the books, the television show takes place in a fantasy world called Westeros, where gods, magic and fantastic creatures exist. The plot is concerned with the lives of different families, who ally, fight and scheme to become the rulers of Westeros. Fig 8 shows the model of the Game of Thrones franchise.

# Defenders of the Triforce

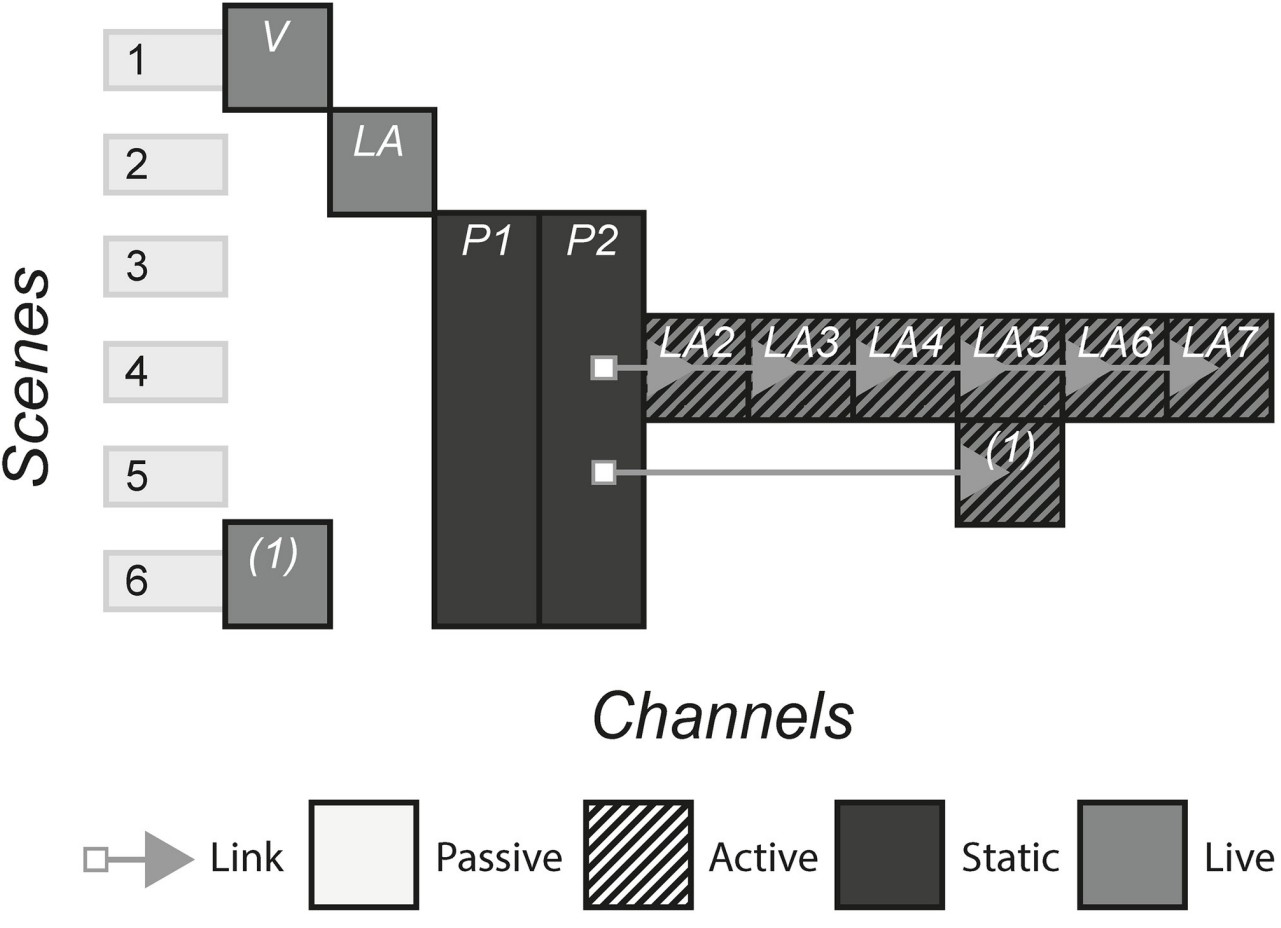

**Fig 5. Defenders of the triforce.**

### 4.8 Harry Potter franchise

Starting with a series of books written by J.K Rowling, the Harry Potter franchise is a collection of media channels that each communicate a storyworld where magic exists, witches and wizards live side by side with non-magical people, known as muggles, and mythical creatures walk the Earth. Fig 9 shows the model of the Harry Potter franchise.

### 4.9 Her Story

Her Story is a game released on the Steam platform that advertises itself as an interactive movie. The game involves the player taking the role of a police officer who has been tasked with investigating the death of a man using a computer. The interface behaves as though you are on a computer, able to view all of the documents, emails and video interviews that are saved on the hard drive. Fig 10 shows the model of Her Story.

### 4.10 Overwatch

Overwatch is a multiplayer first person shooter game. Players battle on large maps with various 'heroes', each with their own abilities, aesthetics and backstory. Overwatch differentiates itself from other shooter games in that the game itself is multiplayer only, with no story mode to communicate any kind of plot. Instead, the storyworld is shown partly inside the game in the form of environment design and character voice lines, and partly via videos, text, comics and other media. Fig 11 shows the model of Overwatch.

### 4.11 Interactive film (Bandersnatch)

Advertised as an 'interactive film' and released on Netflix, Bandersnatch involves the audience watching scenes, then making decisions about what should happen in the story given multiple options. Set in England, the main character work in a video games company who is attempting to adapt the fantasy 'choose-your-own-adventure' books into a video game. Fig 12 shows the model of Bandersnatch.

### 4.12 Pokémon franchise

The Pokémon franchise is a collection of animes, films, games and manga that share the story-world of Pokémon. In this fantasy world, magical creatures with fantastic abilities can evolve

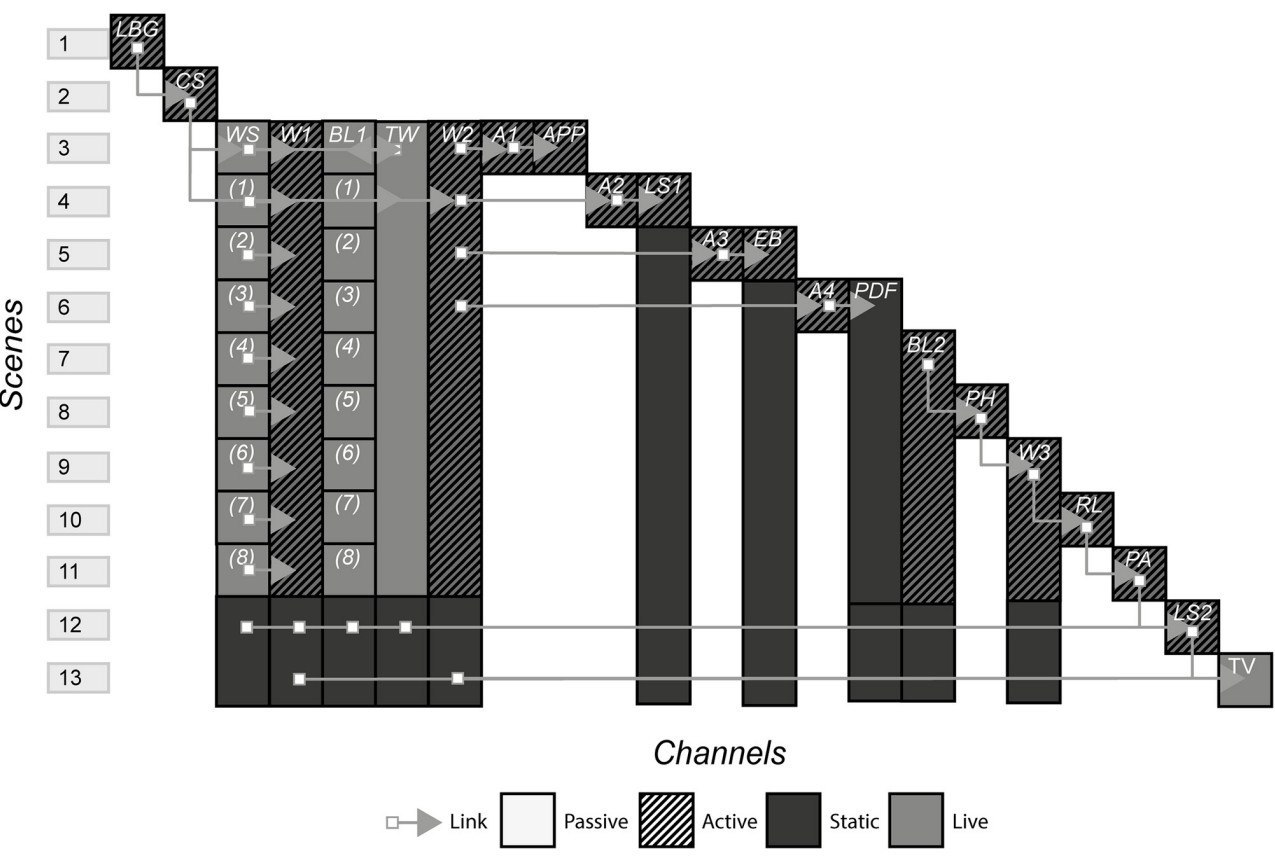

**Fig 6. Dexter ARG.**

# Table Top Role-Playing Game (Example)

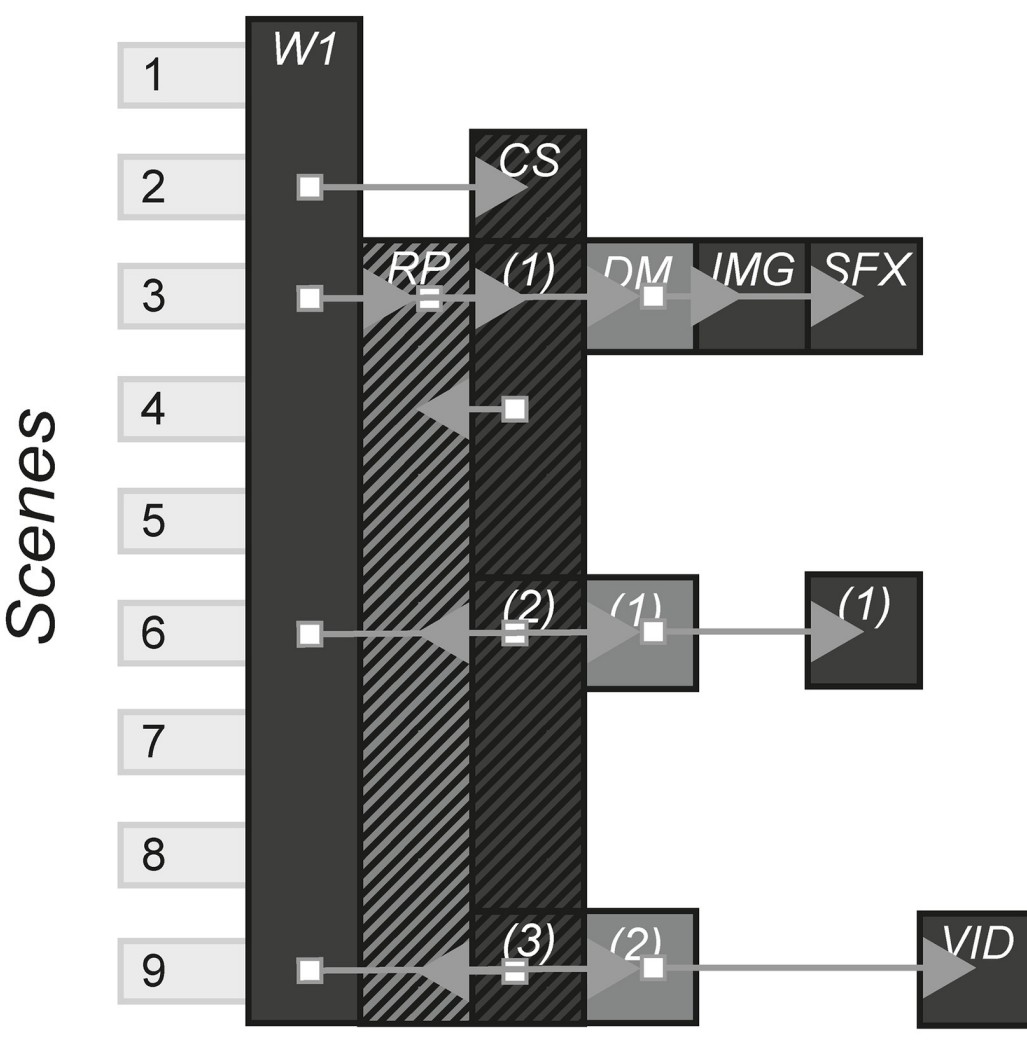

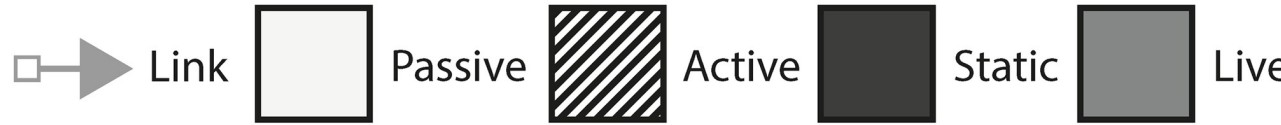

**Fig 7. TTRPG.**

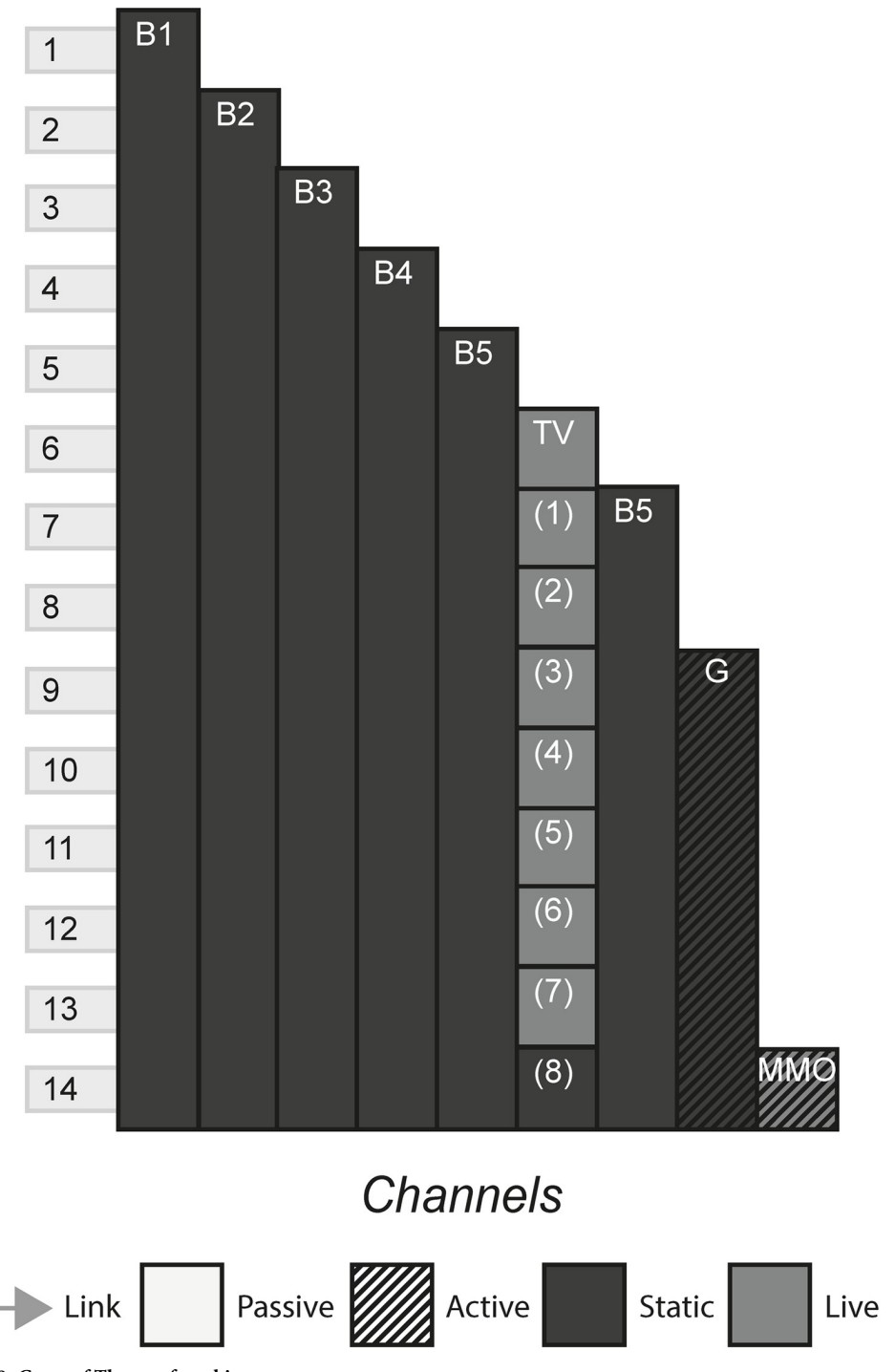

**Fig 8. Game of Thrones franchise.**

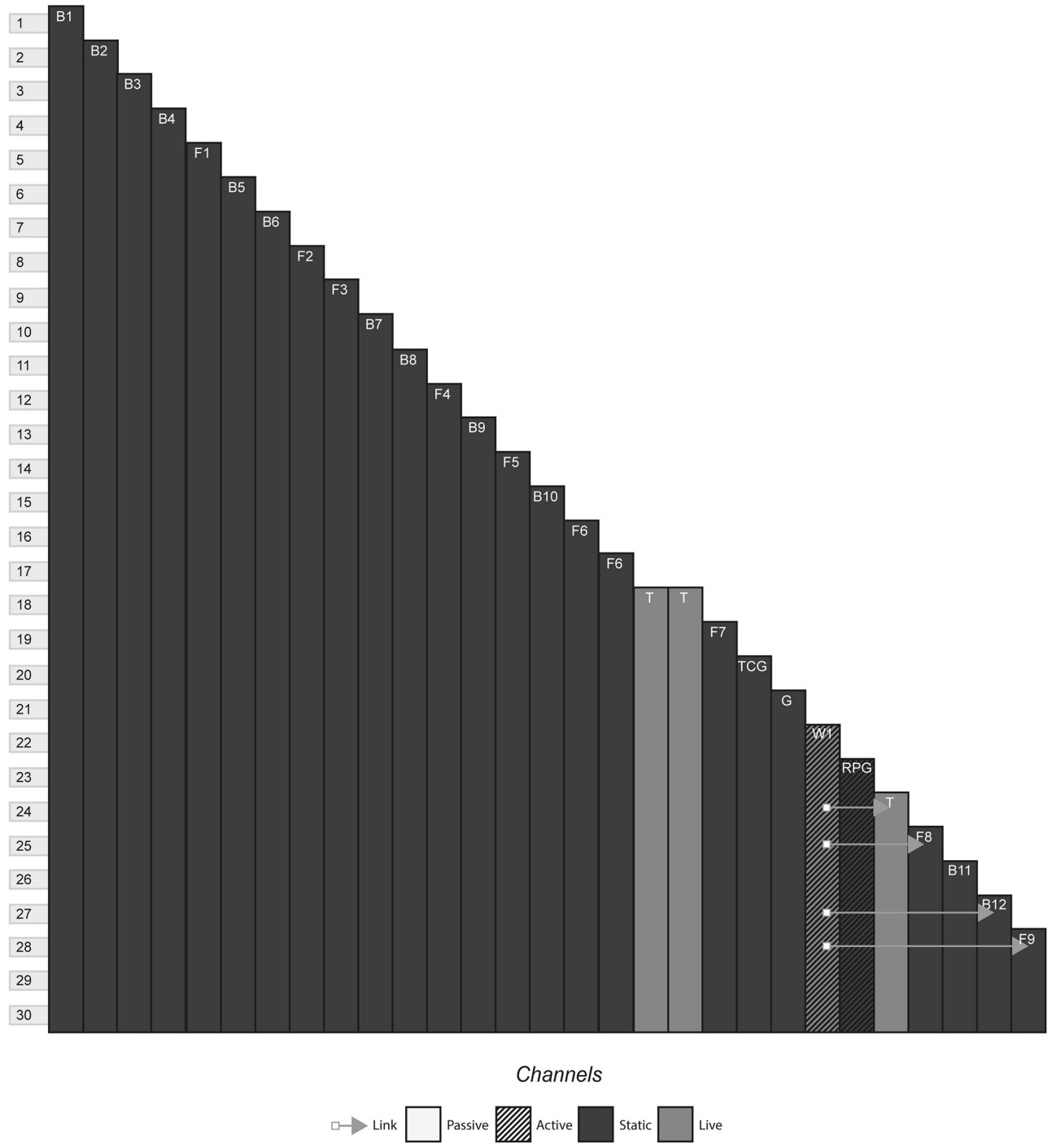

**Fig 9. Harry Potter franchise.**

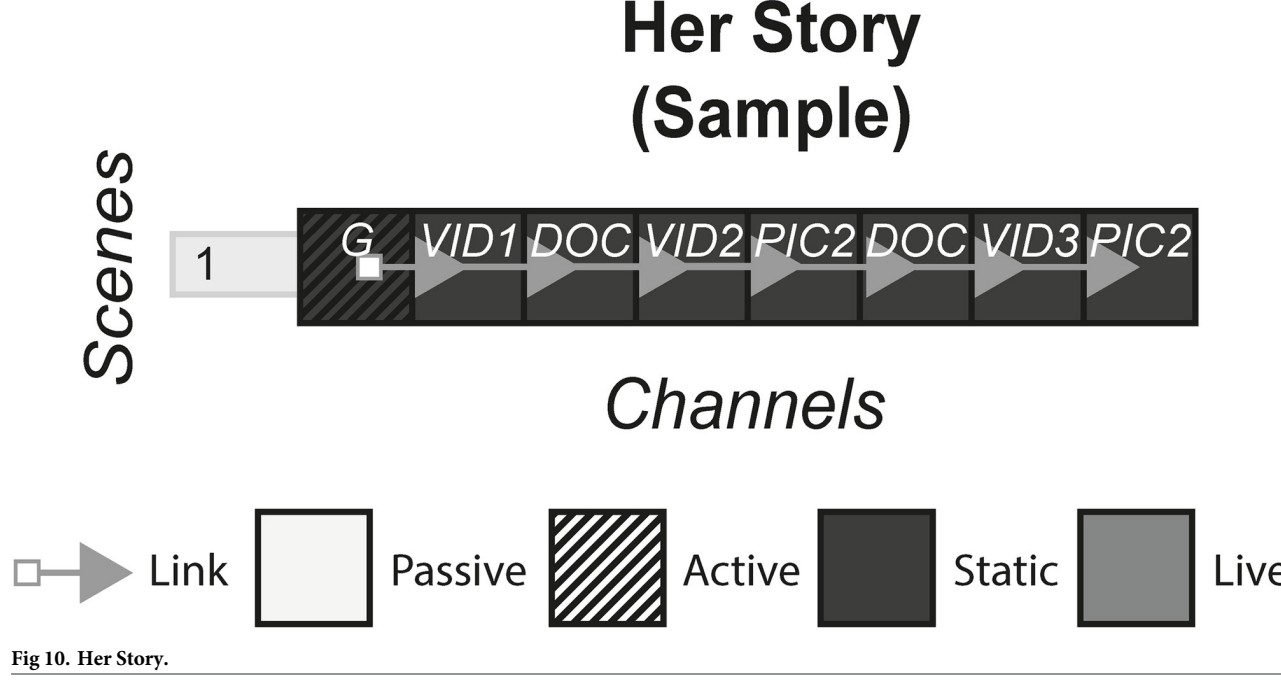

**Fig 10. Her Story.**

instantaneously and are 'caught' by humans, trained and used to battle other trainers Pokémon. However, there are gangs of people who use Pokémon for crime and bad deeds, one of which is known as Team Rocket, who feature dominantly in most of the media. Fig 13 shows the model of the Pokémon franchise.

### 4.13 Pirates of the Caribbean

Starting as a ride at Disney theme parks, Pirates of the Caribbean includes multiple media that tell mythical stories about what the title suggests. The main protagonist, Jack Sparrow, is joined by other pirates and noblemen on various adventures that see them ultimately doing good despite their criminal tendencies. Fig 14 shows the model of the Pirates of the Caribbean franchise.

### 4.13 Prometheus second screen

With the release of the film Prometheus on Blu-ray came a downloadable app, or 'second screen experience' that could be played on iOS and Android devices. The app synced to the movie via audio detection technology, displaying additional content in the form of videos, text, and images that was related to the current scene in the film. The film itself is a prequel to the famous Alien film series. Fig 15 shows the model of Prometheus second screen.

### 4.14 Prometheus campaign

To promote the film Prometheus, a marketing campaign that utilised Web technology was employed by the producers. Beginning as a fictitious TED Talk, the campaign used a number of channels that purported to be real, introducing the main characters and providing a non-essential backstory to the film. Fig 16 shows the model of the Prometheus campaign.

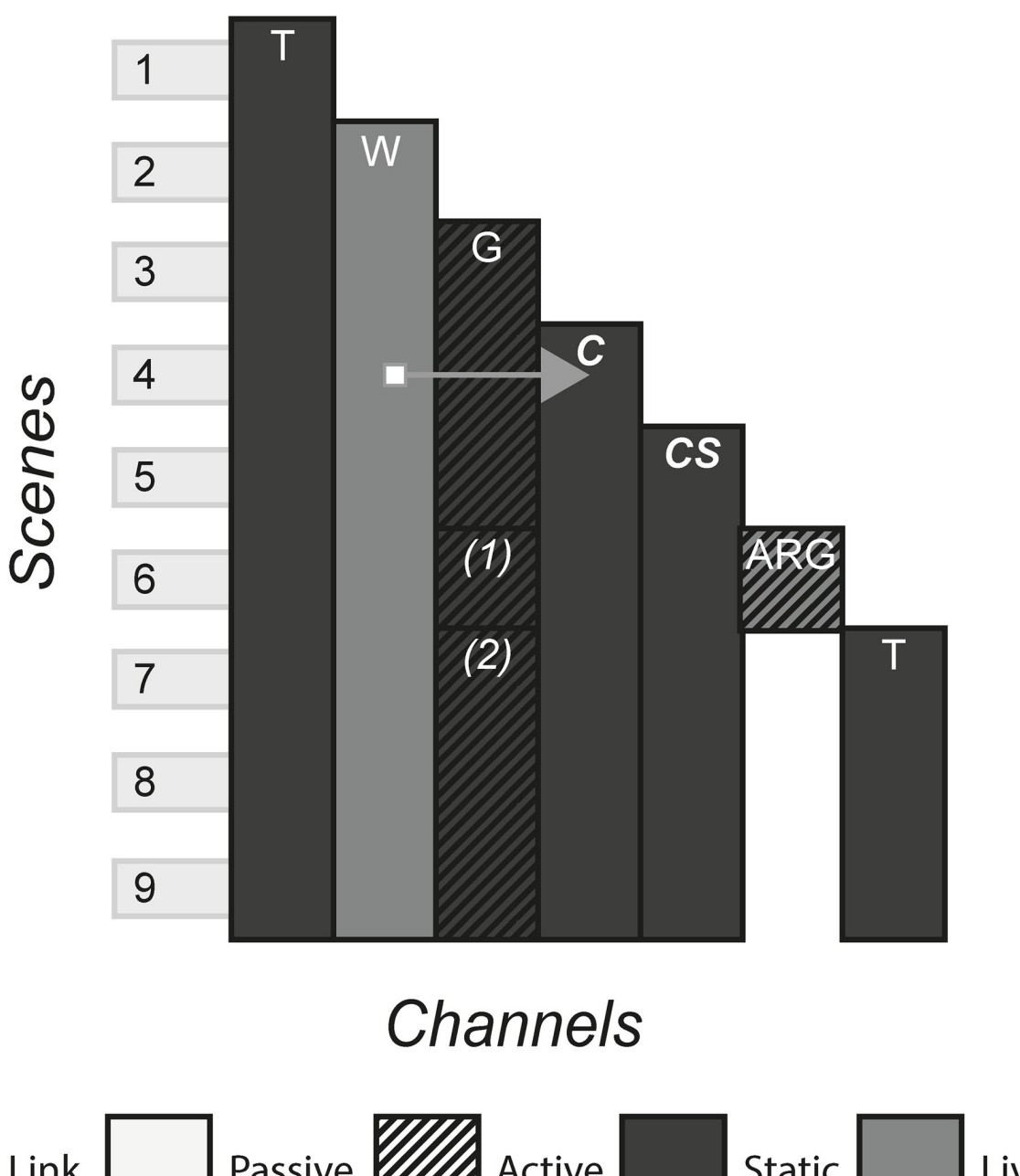

**Fig 11. Overwatch.**

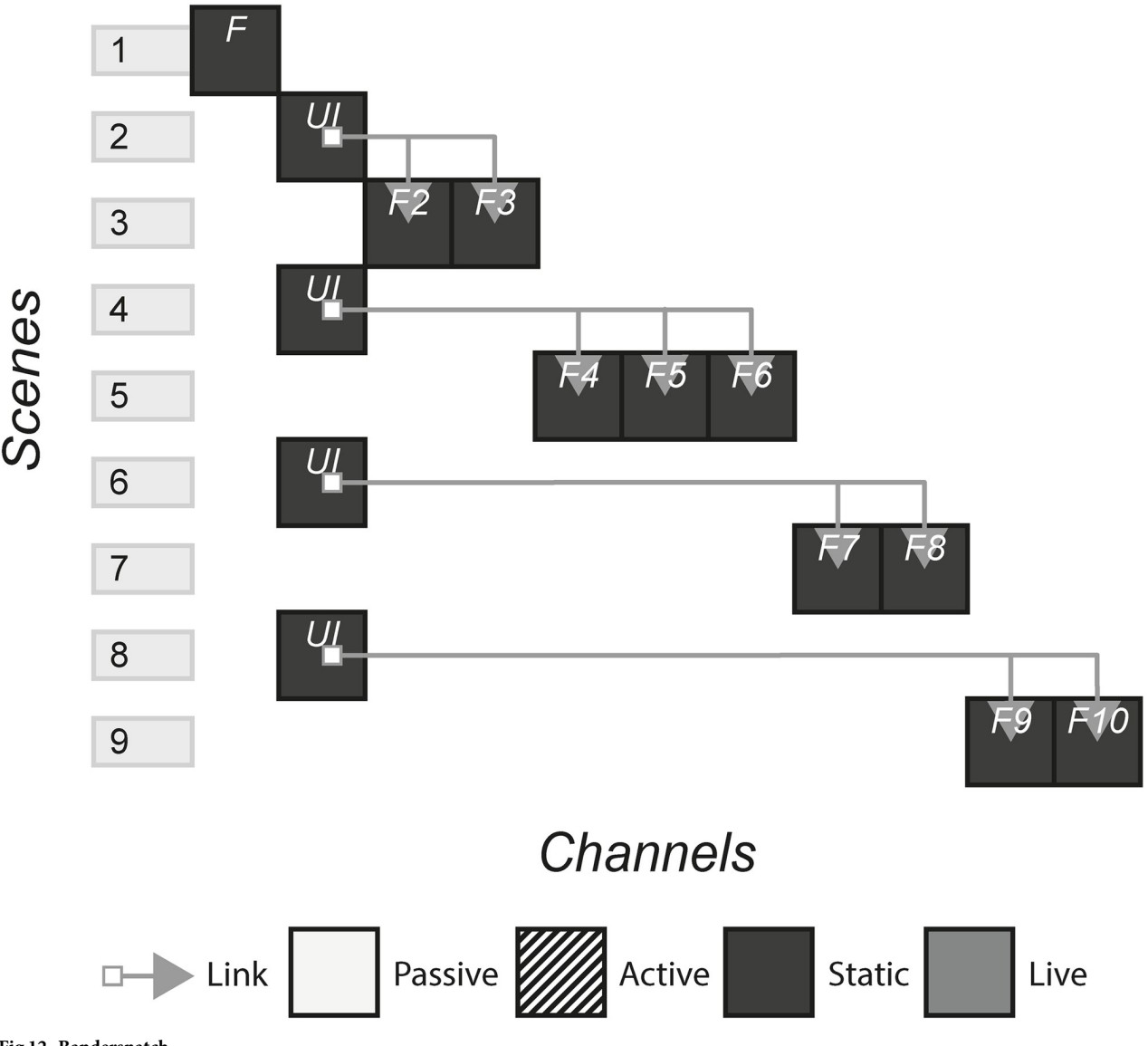

**Fig 12. Bandersnatch.**

### 4.15 Roman Baths

Once the site of a Roman public bathhouse, the Roman Baths located in England is a now a historical landmark that is kept preserved and open to the public to view as an exhibit. The exhibit uses various media to teach visitors about the site, and provide narratives of Roman life including how the baths were made, their religious connotations and what types of people used them. The media used also attempt to create a sense of what it would have been like to visit during Roman times. Fig 17 shows the model of Roman Baths.

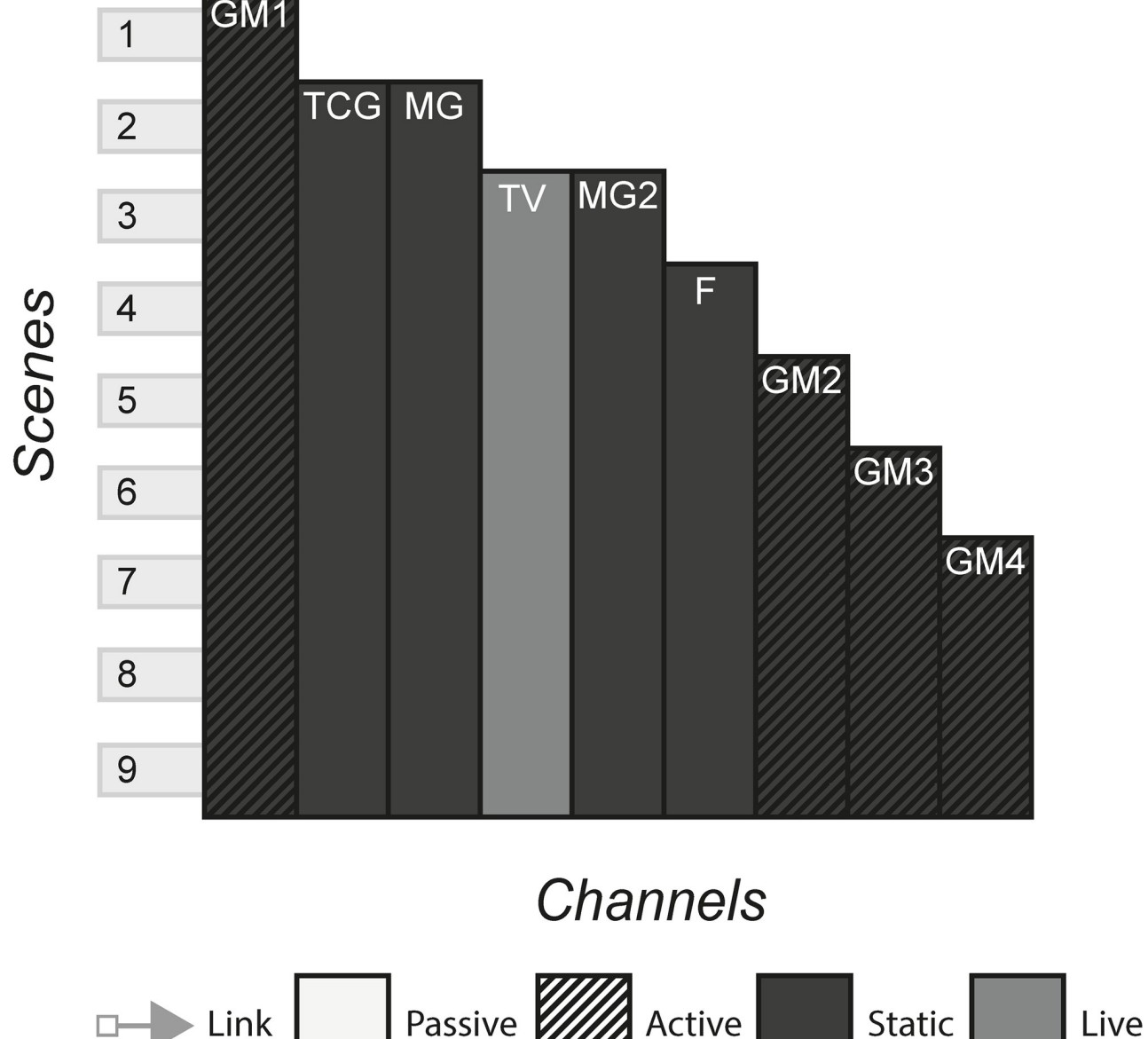

**Fig 13. Pokémon franchise.**

### 4.16 The Black Watchmen

The Black Watchmen is a single player video game released on the Steam platform, that involves the player scanning documents, watching videos, listening to audio files and surfing the Web to find answers that when inputted into the user interface, unlock the next set of

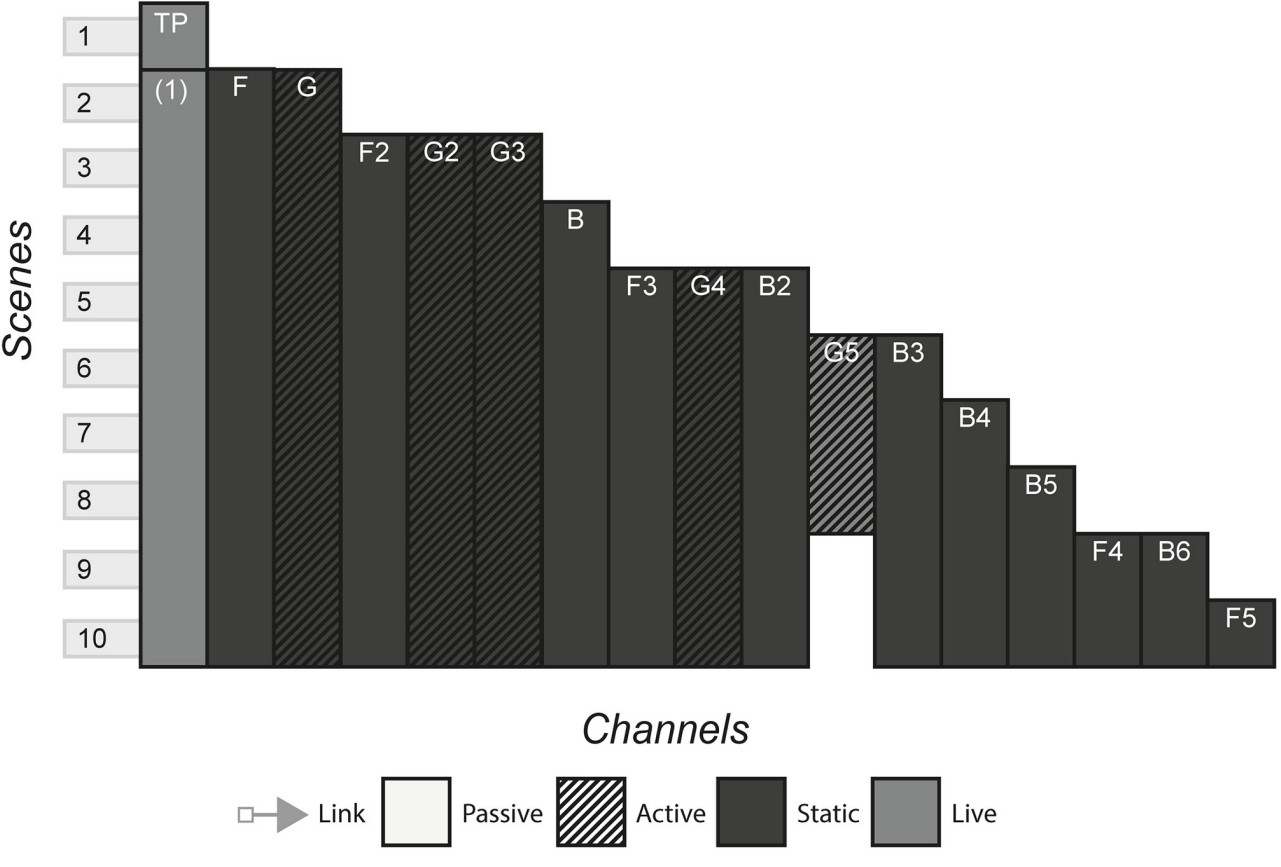

**Fig 14. Pirates of the Caribbean.**

media. The pretext to these puzzles is that the player is a secret agent operative that is tasked with observing these various media channels to find paranormal activity. Fig 18 shows the model of The Black Watchmen.

### 4.17 The Matrix franchise

The Matrix is a collection of media that began with the 1999 release, the Matrix. The film, set in the distant future, is about human-made artificial intelligence that have usurped the human race as the dominant species on Earth. As a way to harvest thermal energy, the AI keep human slaves that are plugged into a machine, unaware of the real world. The creators used different media to tell different stories set inside the world of the Matrix. Although they were each standalone stories, plot threads played out across different media and could only be fully understood if these media were consumed. Fig 19 shows the model of The Matrix franchise.

### 4.18 Westworld season 2 campaign (ARG)

Based on the 1973 film of the same name, Westworld is a television show based in the distant future, when humans have made near-sentient life-like robots. In the show, a company has created a theme park called Westworld, that is filled with these robots that behave as though they are characters from the wild west. People buy tickets to go to Westworld to role-play as

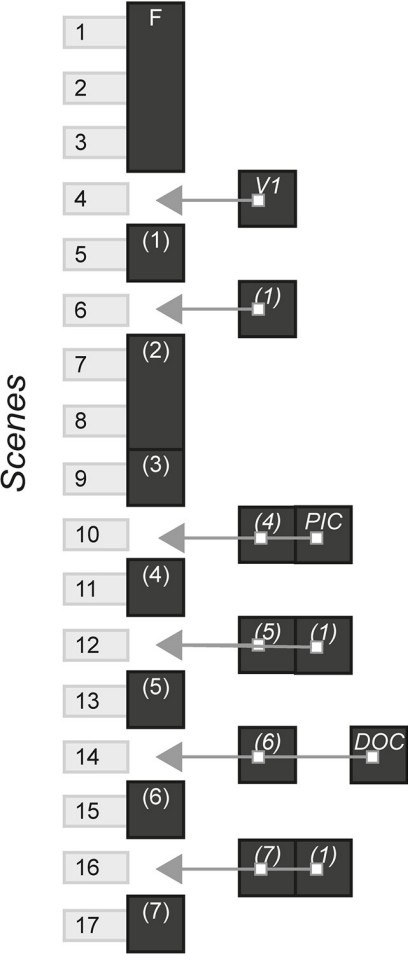

**Fig 15. Prometheus second screen.**

characters, who interact with the robots, go on quests and experience what is known as the ultimate narrative experience. Fig 20 shows the model of the Westworld season 2 campaign.

## 4.19 Why So Serious?

Set in the Batman universe, Why So Serious was an ARG created to promote the upcoming The Dark Knight film. The story involved two characters set to feature in the film, Harvey Dent and The Joker, with the events and actions performed by the players leading to the film. Fig 21 shows the model of Why So Serious?

## 5 Transmedia storytelling patterns

In this section we explore three groups of patterns that have been identified from the case studies; story, navigational and instance. These patterns reflect the possible structural features of any given transmedia story. Story patterns reflect the relationship between a channel and its telling of a story, whether it tells the story by itself, is just part of it or simply enhances it. Navigational patterns relate to how the audiences navigates through the story and their potential channel choice. Instance patterns relate to how the audience interact or consume the instance, different channel's passivity and persistence. The ability to identify these patterns may yield insights into how patterns can be used, which patterns work well with particular types of story and may aid in understanding more about how transmedia genres are formed. With the context of debates that ensure over many of the following terms in mind, we have titled our patterns according to what we saw as the most appropriate description.

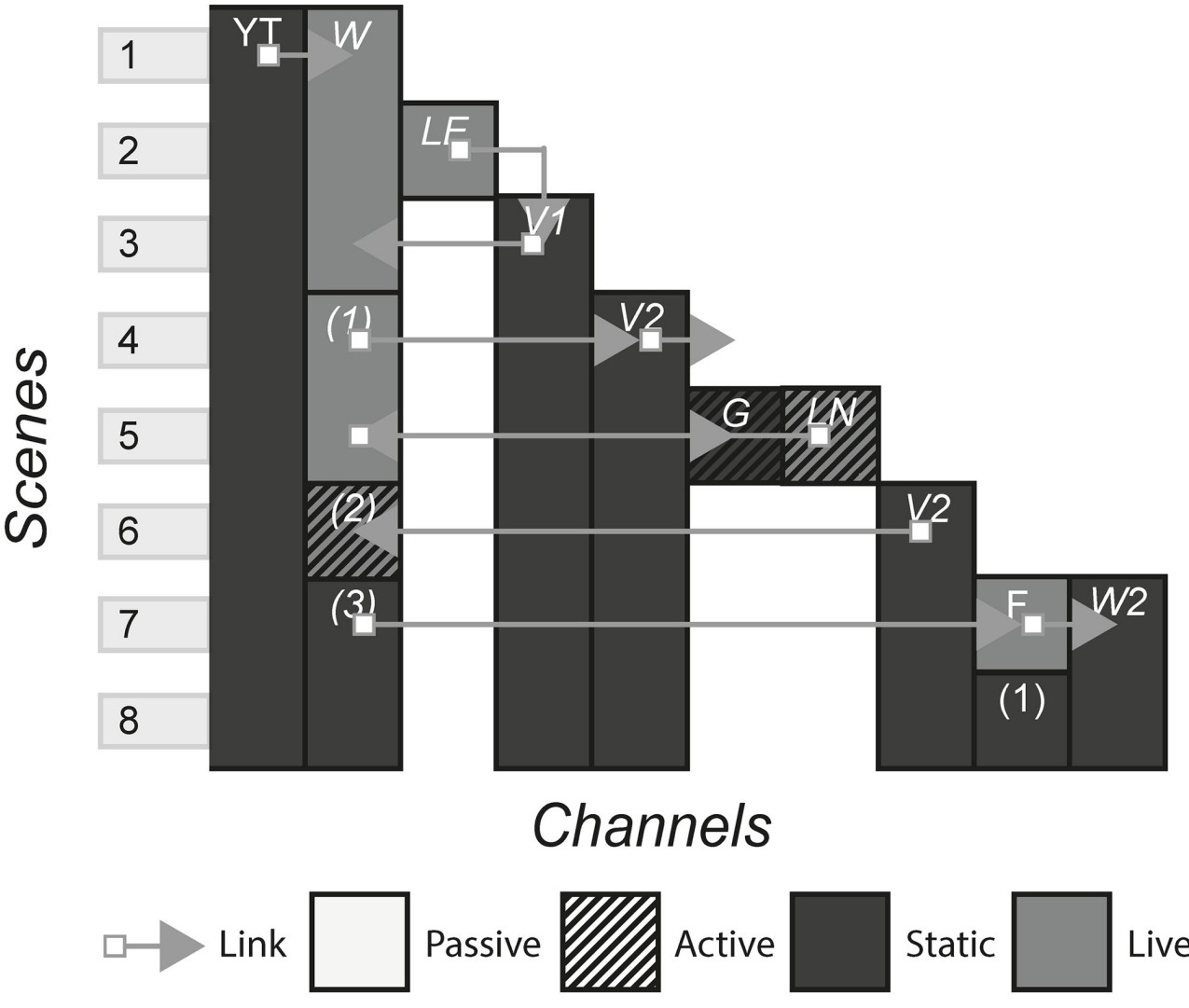

**Fig 16. Prometheus campaign.**

# Museum (Roman Baths)

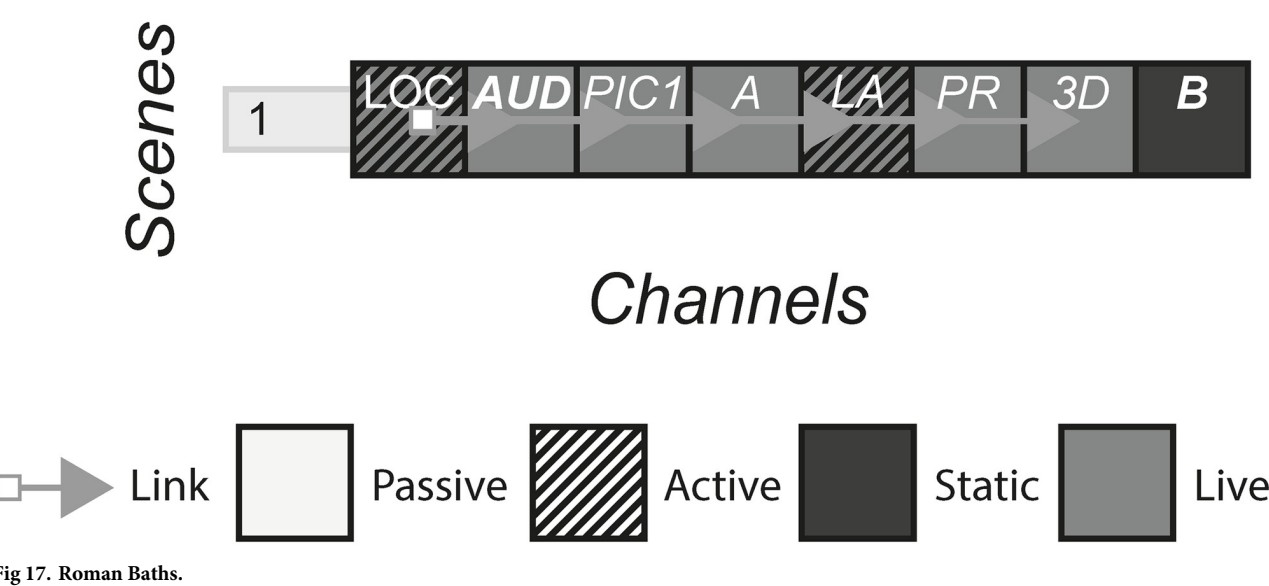

**Fig 17. Roman Baths.**

In this section we describe these patterns, illustrate their canonical form and provide a table that can be used to calculate how strongly a particular pattern is used in a story.

## 5.1 Story patterns

Story patterns relate to the how channels are used to tell the story. We are able to differentiate between transmedia works that use multiple media to tell multiple stories within one fictional world and works that tell only one story using multiple media. We can also extend this to include works where a standalone story is supported by non-standalone multiple media to expand or enhance that story. Table 4 is a summary of the story patterns.

**Many stories.**   The Many Stories pattern reflects how many channels are standalone in relation to the total amount of channels. If a transmedia work uses this pattern, each channel can be experienced on its own in isolation, without requiring the audience to visit other channels. Fig 22 shows an experience with this pattern that includes two films, a game, theatre performance and a book. These channels each have their own story, and make up the storyworld of the experience.

**Portmanteau.**   The Portmanteau pattern shows the total amount of cooperate channels in relation to the total amount of channels. Experiences with this pattern utilise channels that work together to deliver a single story. Each channel, whilst technically can be consumed individually, do not provide a story in and of themselves. Fig 23 shows a portmanteau that tells a single story using two websites, a YouTube video and a characters Facebook account.

**Subsidiary.**   The Subsidiary pattern illustrates the total amount of subsidiary channels compared to the total amount of channels. Subsidiary channels are those that fundamentally rely on another channel, either technologically or via link. Therefore, these channels can't be consumed individually without the channel with which they rely. Fig 24 shows a Blu-ray that links to an app that is reliant on the Blu-ray. An example of this could be apps that use audio

# The Black Watchmen

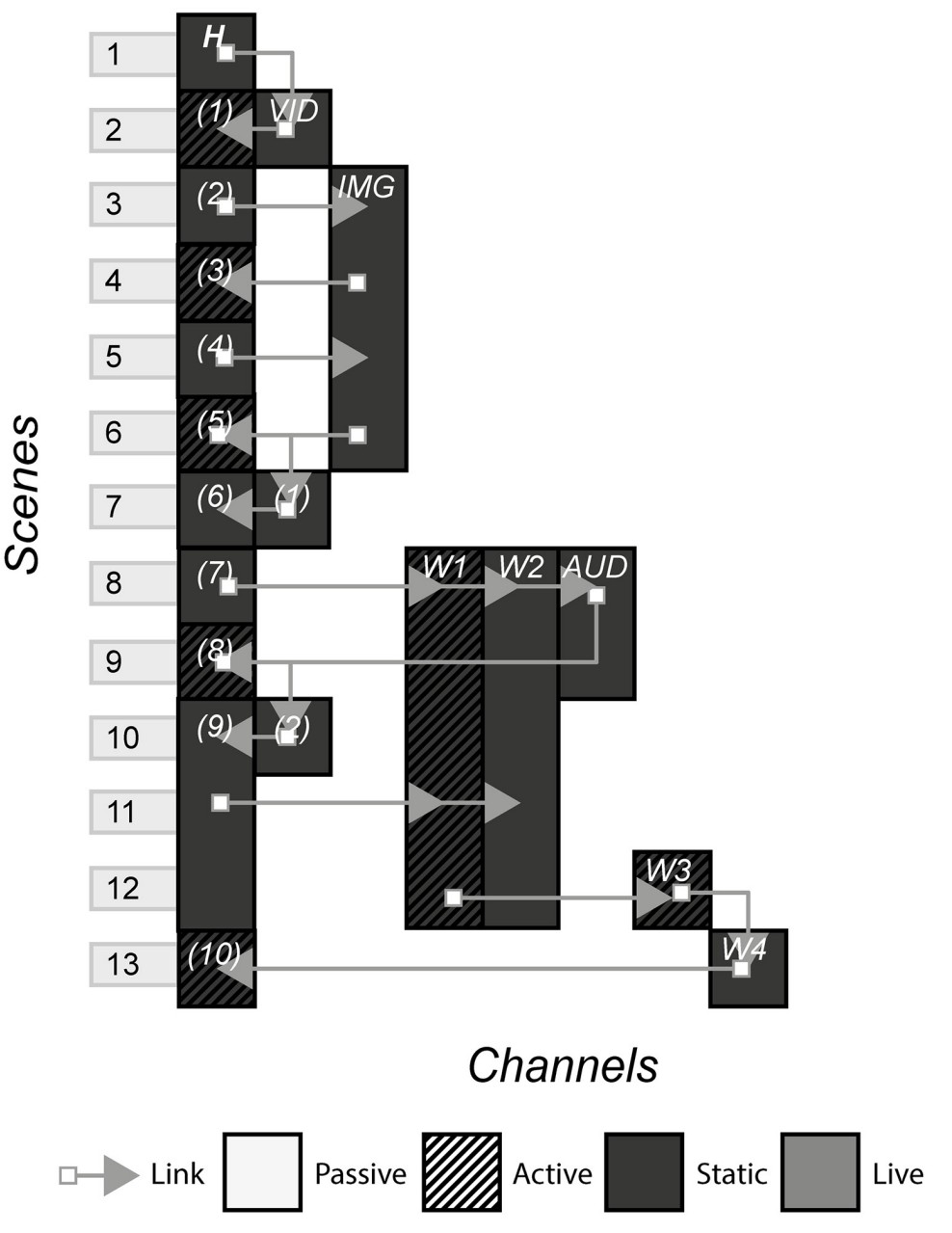

**Fig 18. The Black Watchmen.**

sync functionality that technologically require the film to be playing in order for the content to emerge.

## 5.2 Navigational patterns

Navigational patterns relate to how an audience member navigates their way through the story as new content releases or is made available to them. We are able to differentiate between different ways an audience member is expected to experience a story, whether going from one

# The Matrix Franchise

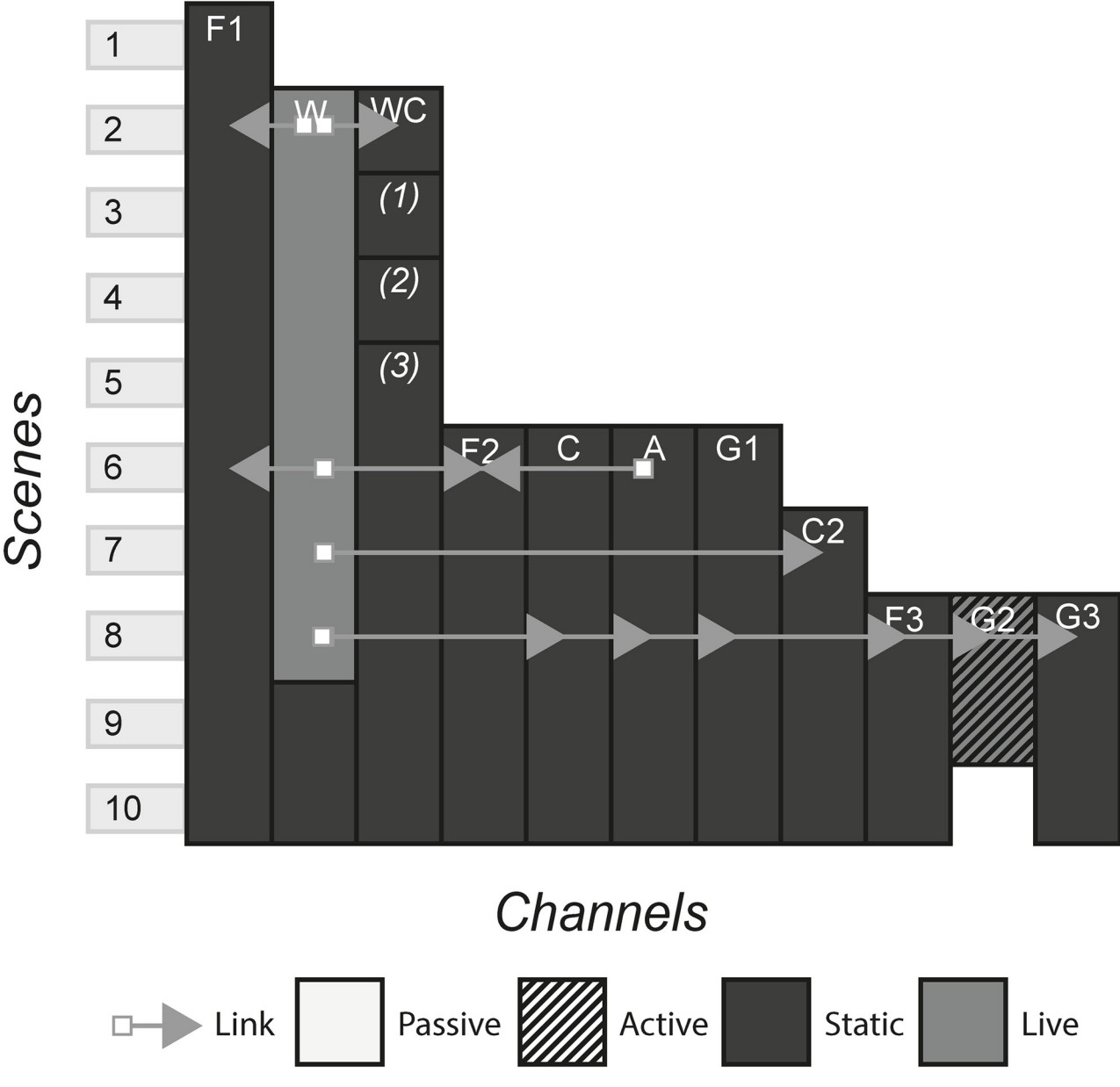

**Fig 19. The Matrix franchise.**

channel to the next sequentially, having multiple options or a combination of the two. Table 5 is a summary of the navigational patterns.

**Linear.** The Linear pattern reflects how many scenes include only one instance, relative to the total scenes. Linear experiences are those that do not give the audience any channels options. Typically, the audience starts on one channel, then moves on to the next and so on, without the ability to go back and see previous content unless the experience is restarted. Fig 25 shows a story that plays out on five HTML websites, which each scene containing a different website, and without the ability of going back to a previous website.

**Fig 20. Westworld season 2 campaign.**

**Non linear.**   The Non-Linear pattern represents how many scenes have multiple new instances compared to the total scenes. If a scene allows the audience to choose which channel they go to next, the experience intends for the audience to make a choice as to the order in which they experience the channels. Fig 26 shows a story dispersed across six websites, where each scene gives the player a choice as to which instance is experienced first.

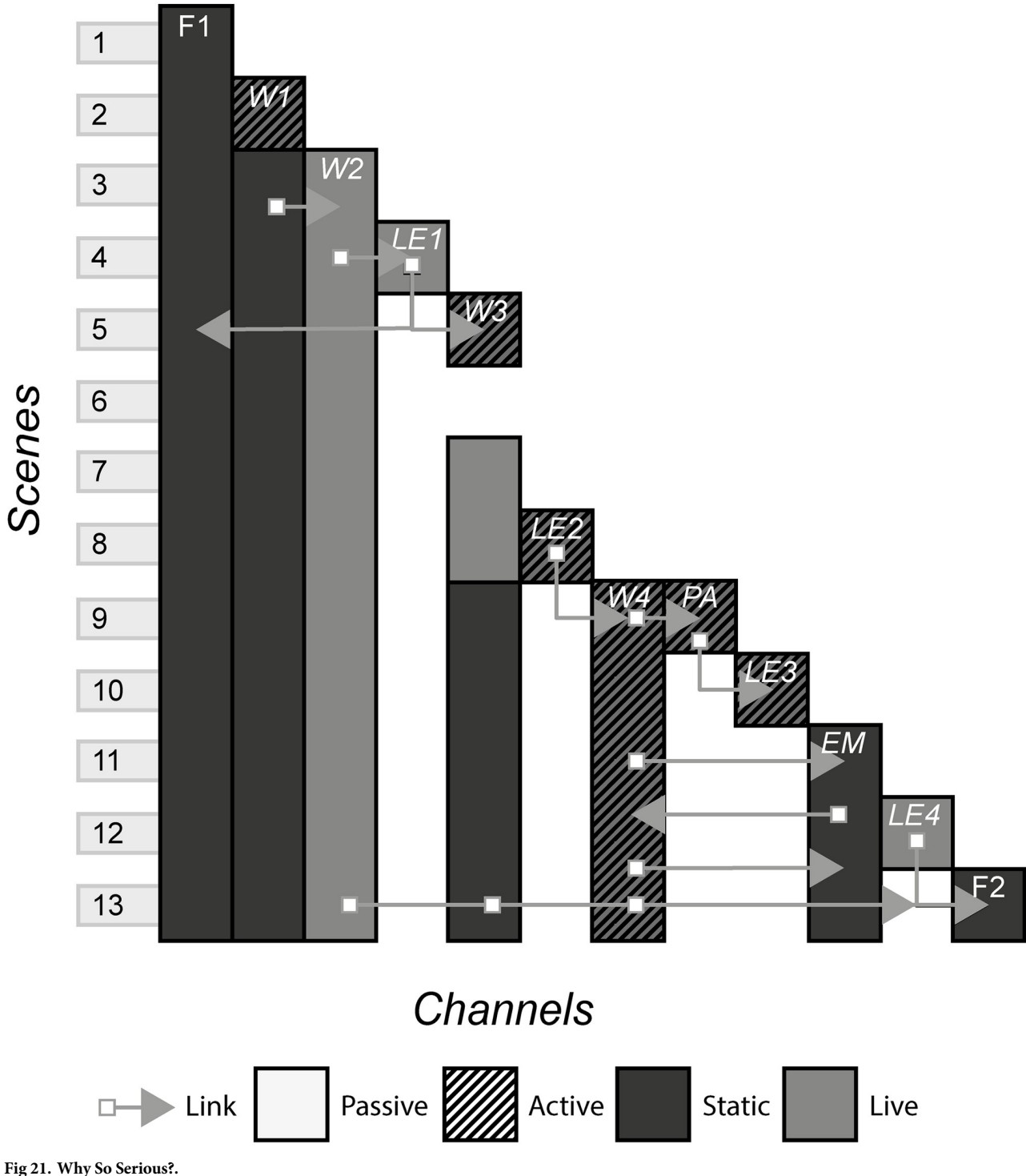

**Fig 21. Why So Serious?.**

**Table 4. Story patterns summary.**

| Pattern | Metric |
|---|---|
| Many Stories | High number of standalone channels |
| Portmanteau | High number of co-operative channels |
| Subsidiary | High number of subsidiary channels |

**Cumulative.** The Cumulative pattern shows how many scenes includes one new instance, but with the ability to go back to old instances, compared to the total scenes. A cumulative experience is a mix between the two former patterns, producing a linear effect with regard to new content, but allow choices to be made in whether to access old content and reinterpret information in light of the new instance. Fig 27 shows an experience where in scenes 2–5, a new instance is available but the player is also able to go back to the first instance in scene 1.

**Connected.** The connected pattern is determined by the amount of unique links each channel has compared to the total amount of channels. If a channel (A) instances have several

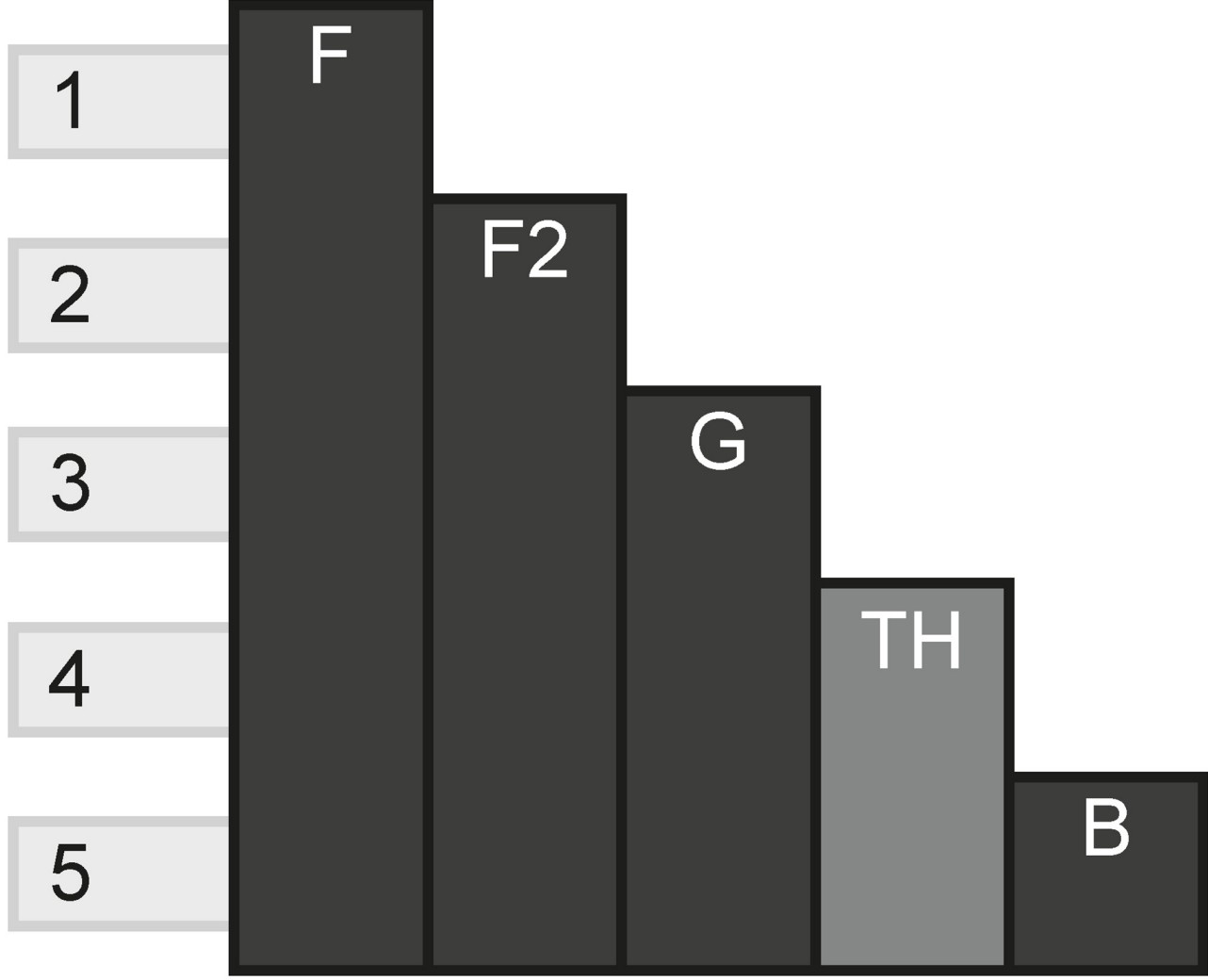

**Fig 22. Many stories.**

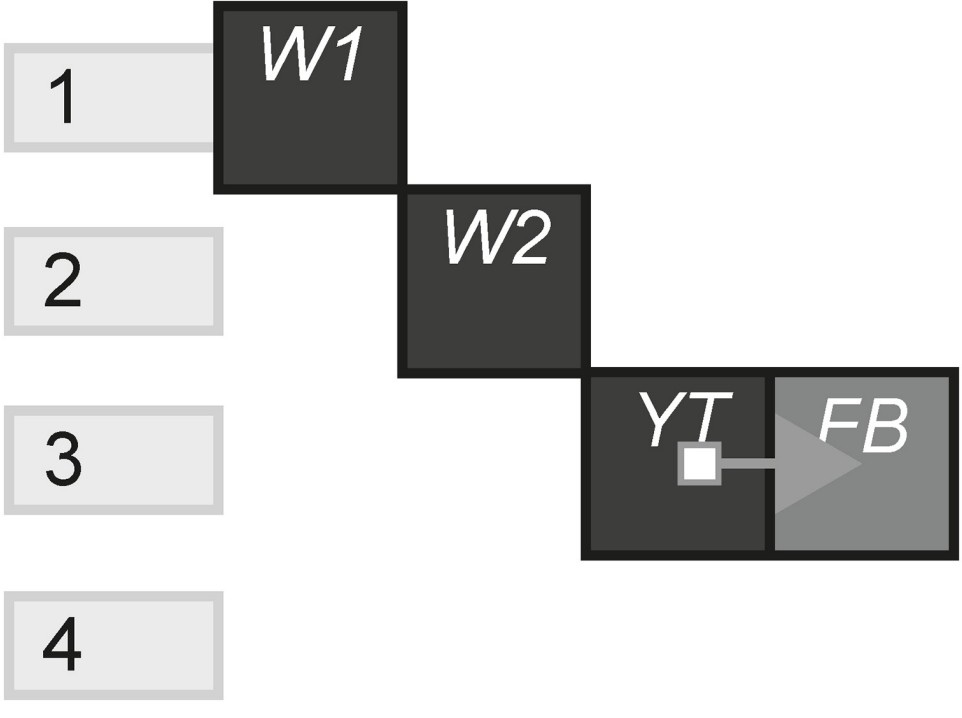

**Fig 23. Portmanteau.**

links to the same channel (B), then there is only one unique link (A to B). Fig 28 shows all channels connected to each other.

## 5.3 Instance patterns

Instance patterns relate to how the audience interact or consume the instance. We can differentiate between experiences that facilitate role play, are consumed passively, have to be experienced at a specific place and time and those that can be accessed at any time after they are released. Table 6 is a summary of the instance patterns.

**Role-play.** The Role-play pattern illustrates the total number of scenes that have active instances compared to the total amount of scenes. Fig 29 shows an experience that has at least one active instance per scene, meaning the experience relies highly on the use of role-play.

**Audience-centric.** The Audience-Centric pattern is the opposite to the Role-play pattern, denoting experiences that deliver their content to audiences in a passive way for them to consume, without being part of the story or role-playing any pre-made characters. Fig 30 shows a story with no active engagement.

**Live event.** The Live Event pattern is determined by the amount of scenes that have a live instance compared to the total amount of scenes. Experiences with this pattern tend to be ephemeral (e.g. a music gig), or have to be consumed at a specific time and place (e.g. a theatre performance or theme park). Fig 31 shows a live experience.

**Artefact.** The Artefact pattern is the opposite of the Live Event pattern, and is determined by the amount of scenes that have a static instance compared to the total amount of scenes. Experiences with this pattern are ones that are consumed by the audience any no particular time. The experience can be played, replayed and paused. Depending on the particular channel, static instances can be consumed in many places (e.g. websites on phones, tablets and computers). Fig 32 shows an artefact experience.

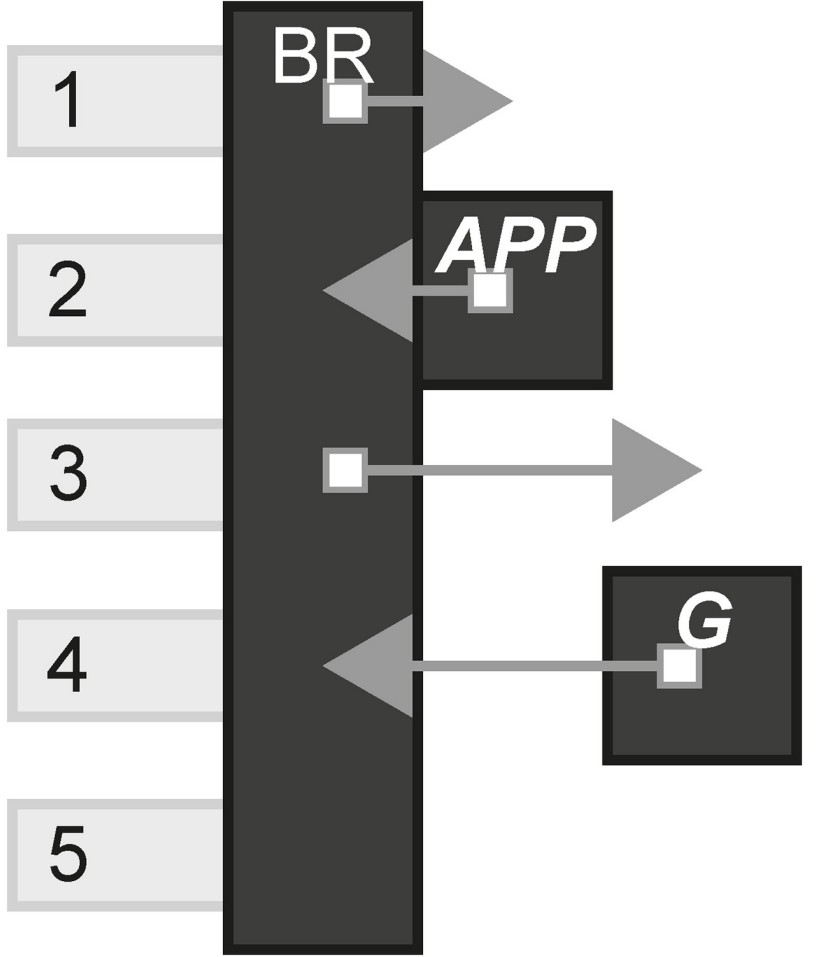

**Fig 24. Subsidiary.**

## 6 Forms of transmedia storytelling and their use of patterns

In this section we discuss how patterns are used in five forms of transmedia storytelling; interactive films/second screen apps, ARGs, media franchises, escape rooms, table-top role-playing games and exhibits. We explore how these forms illustrate the patterns in different ways using examples from the case studies. Table 7 shows the primary patterns of each form identified from the case studies.

### 6.1 Interactive films/second screen

**(APP, bandsnatch, prometheus, Her Story).** This form of story includes films that have elements of interaction with the audience in various ways. In a standard film, the audience

**Table 5. Navigational patterns summary.**

| Pattern | Metric |
|---|---|
| Linear | High number of scenes with one instance. |
| Non Linear | High number of scenes with multiple new instances |
| Cumulative | High number of scenes with only one new instance |
| Connected | High number of links per channel |

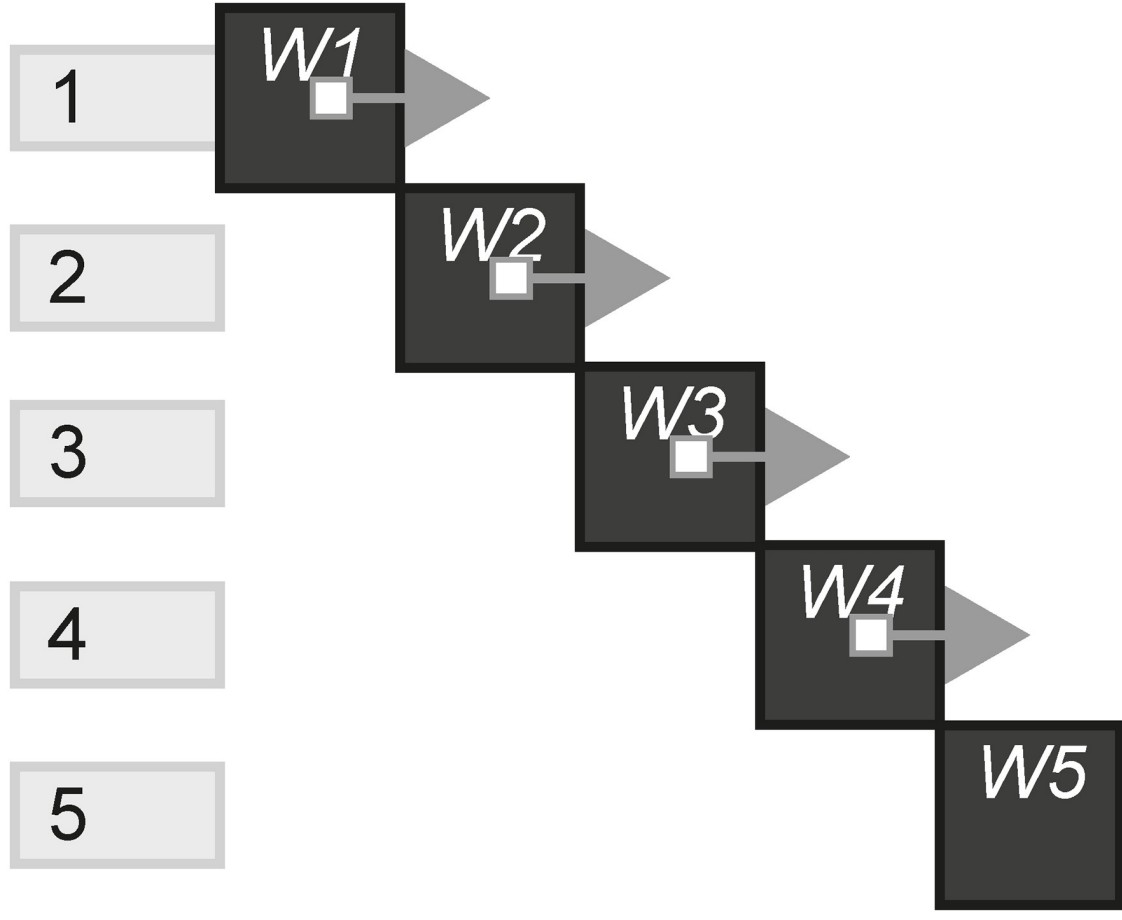

**Fig 25. Linear.**

passively consume the content either live e.g. a cinema or statically e.g. a Blu-ray, by watching a screen and listening to the music and sound. In interactive films and second screens, the audience is afforded an extra duty in some way, either through making a choice or consuming extra content.

**Story patterns.** This form uses the Subsidiary and Portmanteau pattern. In APP and Prometheus, an app is downloaded onto a device which uses audio sync technology or Wi-Fi to deliver subsidiary channels that display different channels depending on the specific scene of the film. In Bandersnatch and Her Story, each channel works together to deliver the story.

**Navigational patterns.** This form uses the Non-Linear form most commonly, with the experiences allowing the player to choose from a selection of channels which ones they want to view at any given time. In APP, when there is a bomb under a café table near the characters, a timer is shown on the second screen app, letting the audience know exactly how long is left before the bomb goes off. Here, there is a connection to what is happening on the screen as the audience is given an extra sense of impending doom. They can either watch the film, look at their phone or both. However sometimes the form may utilise the Linear pattern by controlling what channel the audience is viewing. In Prometheus, when the audience see an old Mr Weyland for the first time in the film, the app pops up with a TEDx style video of Mr Weyland in his younger days, giving the audience a sense of who he is and what his motivations are whilst the film pauses. In Prometheus, the audience is not getting an extra sense of something

we know is going on in the scene like in APP, but they are getting extra information that may or may not affect how they view Mr Weyland for the rest of the film.

For Non-Linear stories such as Bandersnatch, that lock out all other options once one is picked, the story that is experienced could be different between different audience members. The story can be either one type of "play through" or could be every single possibility. Non-Linear stories that do not lock options, but allow the audience to view all channels, are more likely to fall under the latter defined story, even though the order in which these channels are consumed could change the reading of subsequent channels.

The Connected pattern is sometimes used in this form to push the audience from a second screen app back to the film at appropriate times e.g. Prometheus, but other times there are no links and the audience is left making their own decision on where to direct their attention e.g. APP.

**Instance patterns.**   Both the Live and Artefact pattern are used in this form, with each pattern having an impact on the use of other patterns. For example, stories that use the Non-Linear and Live pattern together may risk cognitive overload, breaking the immersion, and disrupting the pacing set by the film. Artefact stories may pause the film in order for the audience to take their time with all the content.

It is difficult to see how this form could use the Role-play pattern, and our case studies illustrate the exclusivity of the Audience-centric pattern. One reason for this is the pre-rendered nature of film, with little ability to allow for role-play.

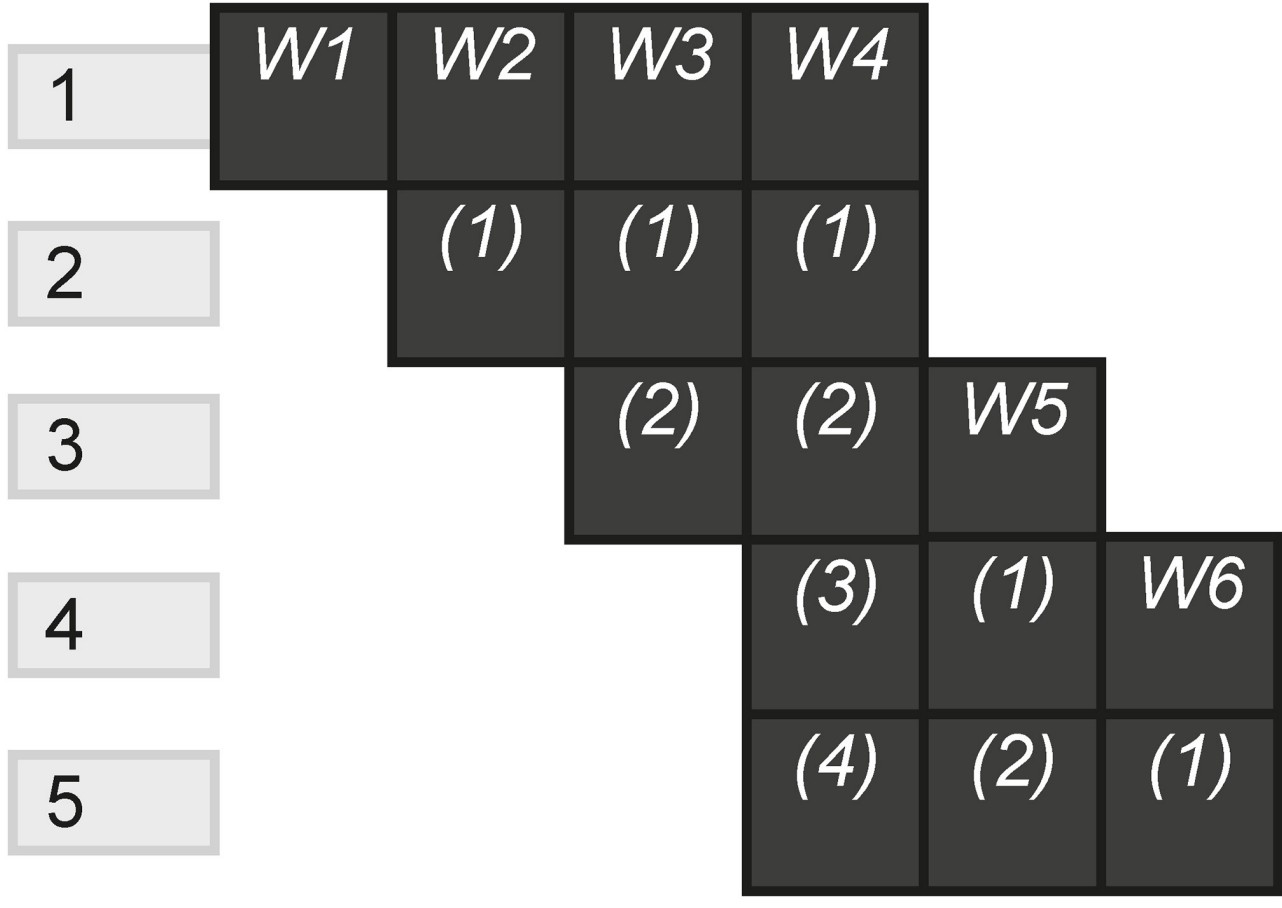

**Fig 26. Non linear.**

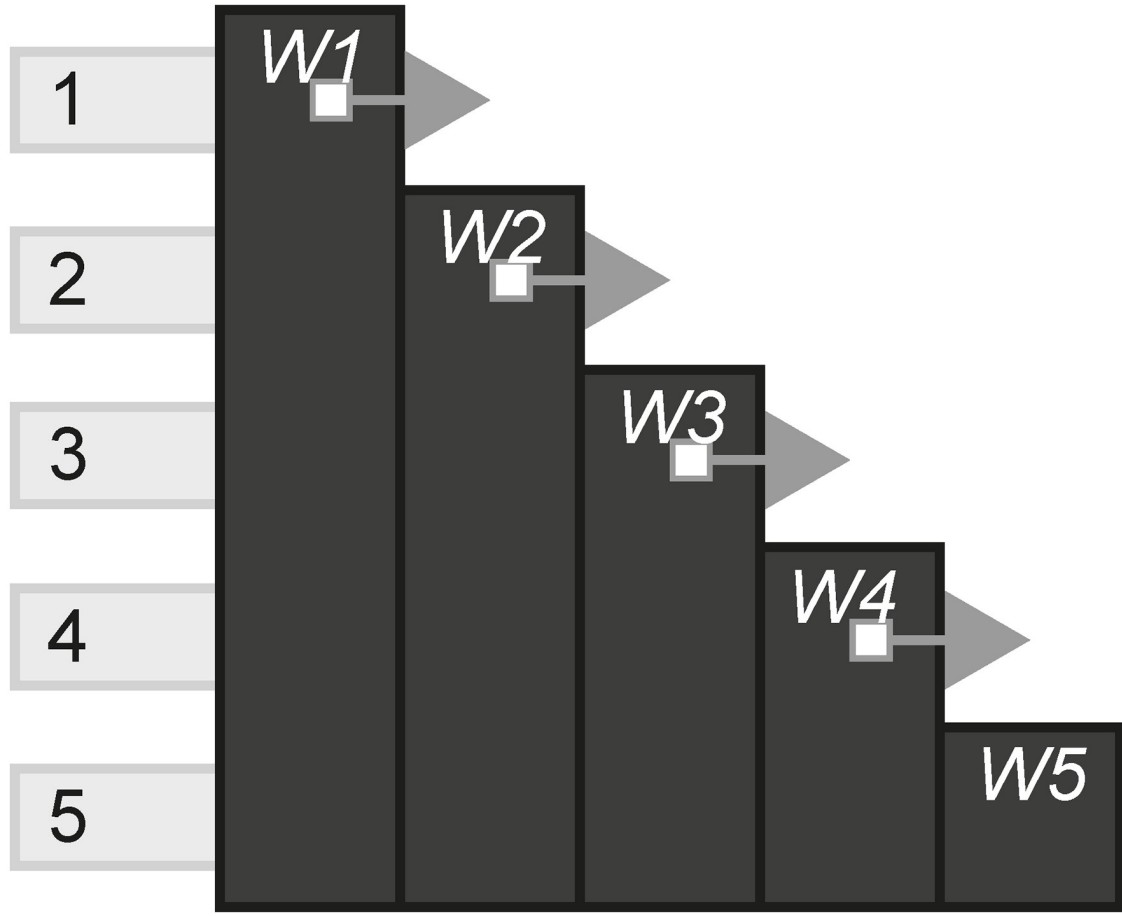

**Fig 27. Cumulative.**

## 6.2 ARGs

**(19 Reinos, Prometheus, Westworld, Dexter, Why So Serious?, The Black Watchman).** ARGs, often used as a marketing tool, are defined by their distinctive trait of blurring reality with fiction by using multiple channels that purport to be authentic e.g. fictitious company websites, character social media pages and character blogs. Audiences tend to be large and collectively hunt down clues from these channels to find additional channels and piece together a story that in many cases provides backstory and context to a standalone channel.

**Story patterns.** All ARGs use the Portmanteau pattern in that most channels are not self-contained and only make up part of the whole experience. With ARGs that are part of a larger franchise, the Many Stories pattern appears and can be considered to be part of the ARG, especially when an ARG continues to run after or during the standalone channel release. In Westworld and 19 Reinos, channels were producing new instances with each new episode, incorporating the standalone channel into the experience. On the other hand, Dexter finished before the television show had aired its first episode, and did not continue after.

**Navigational patterns.** The navigational patterns used by ARGs is predominantly Cumulative, with some Non-Linear. Producers, or "puppet masters" as they are known, will create Non-Linear scenes when they want the story to be pieced together by the audience in an ambiguous way. When they want the audience to get a piece of the story in the same way, they

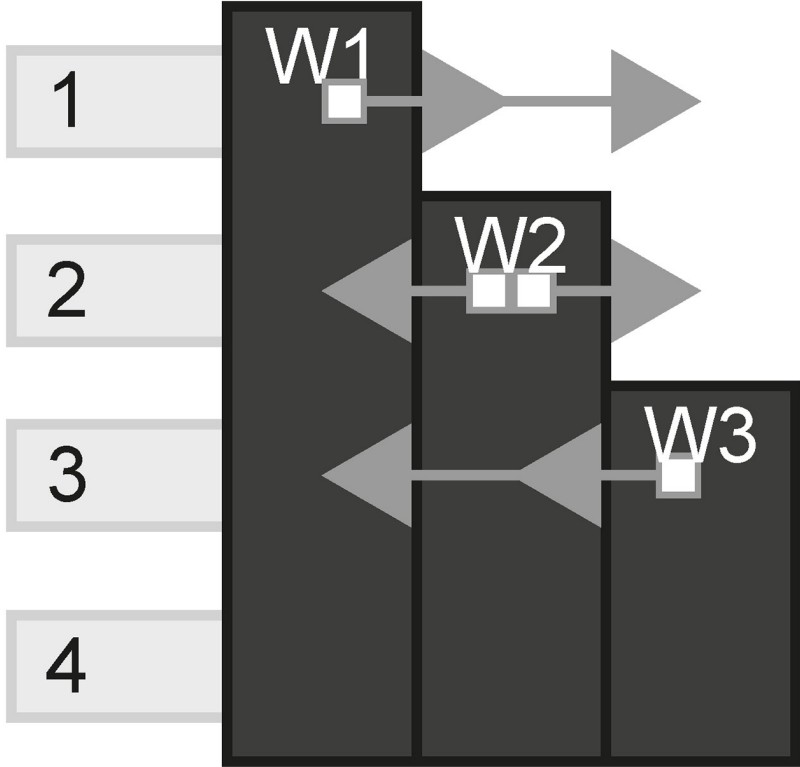

**Fig 28. Connected.**

may switch to Cumulative to restrict the audience to one new instance on one particular channel. This produces the effect of firstly causing the audience to flock to one channel, and then potentially pushing them back to older content to reinterpret given the new update e.g. a Cumulative scene gives the company password that allows access to content on their website that was inaccessible in previous scenes. The Non-Linear and Cumulative pattern can however create different responses to the story in different audience members, especially when there is a standalone channel. In Westworld, those who visited the company website and saw secret videos of James Delos before watching the episode where they revealed he was in fact a robot, would have a different response to those who did not see the videos and suspected he was a human.

ARGs scarcely use the Linear pattern, because it disrupts the feeling that what is happening in an ARG is real, by limiting the events of the fictional world by allowing access to only one instance per scene. The Linear pattern also damages the massive multiplayer infrastructure that many may argue is the most appealing trait of ARGs by not allowing the excitement of different people from around the world finding different instances at the same time, pooling their

**Table 6. Instance patterns summary.**

| Pattern | Metric |
| --- | --- |
| Role-play | High amount of scenes with active instances |
| Audience-centric | High amount of scenes with passive instances |
| Live Event | High amount of scenes with live instances |
| Artefact | High amount of scenes with static instances |

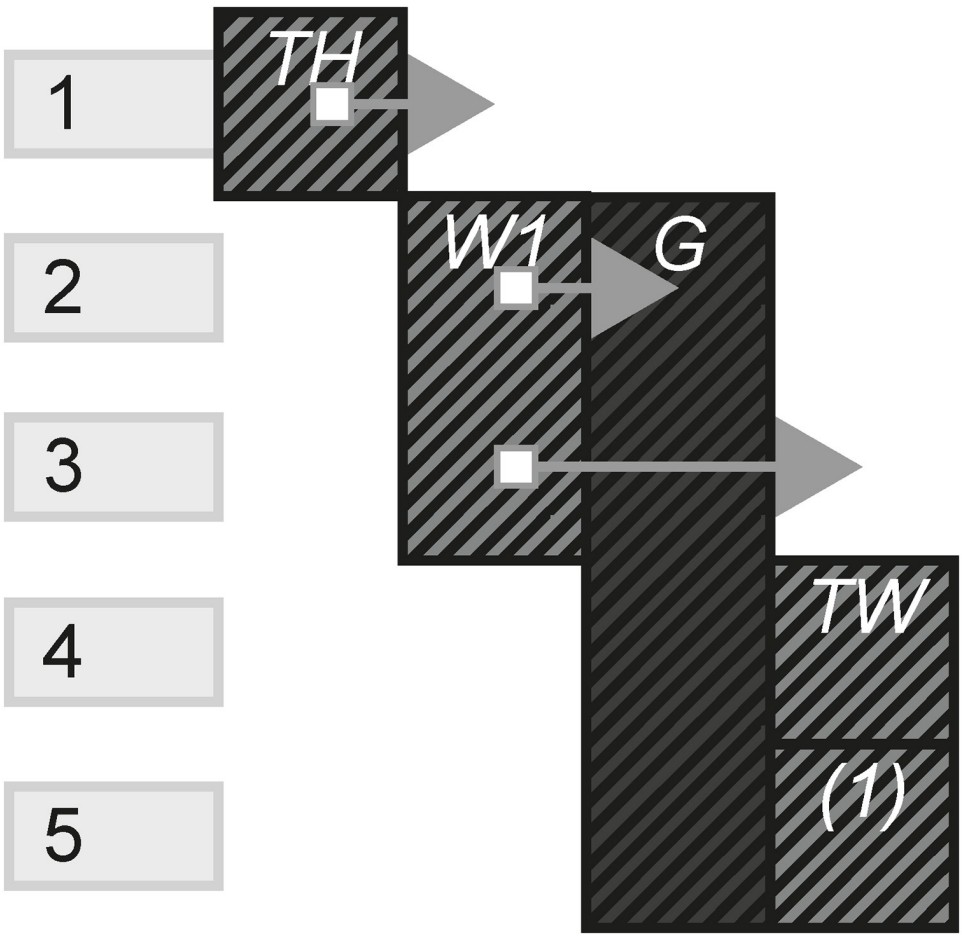

**Fig 29. Role-play.**

collective intelligence, solving the puzzles and hunting down further channels. However, ARGs such as The Black Watchmen that advertise themselves as single player ARGs, are better suited to having the Linear Pattern. Here, a single audience member can get some of the characteristics of a conventional ARG such as cryptic clues, fragmented story and multiple channels without having to interact with other people. The consequence is that puzzles, channels and to some degree the level of complexity of the story, have to be carefully constructed to accommodate the abilities and comprehension of a single person. In The Black Watchmen, a mission can consist of watching a video, examining a document for clues and inputting a keyword into an interface. The channels of a single player ARG are also limited, e.g. the live crime scene set up in Dexter would not have been made for just one person for economical and practical reasons. The trade-off is that unlike Non-Linear/Cumulative patterns, the producers have much more control over what the audience consume and are more likely to get a common reaction and emotional response amongst all audience members. Taking our previous example from Westworld, in a Linear scenario, the producers could decide whether they want the audience to suspect that James Delos is a robot, and every audience member would be given the same information at the same time.

ARGs showcase the Connected pattern the most out of all forms, with many clues scattered across the various channels, that interlink many of them. These links are known as "rabbit

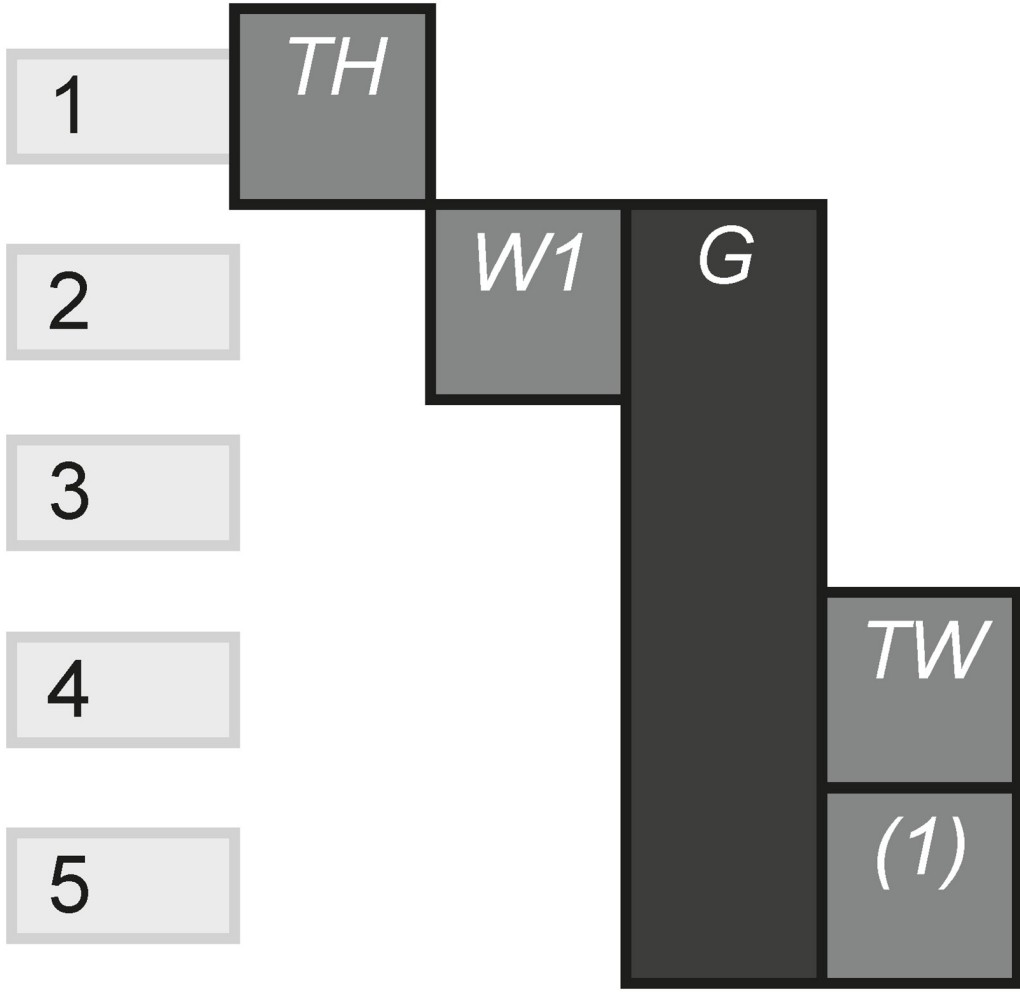

**Fig 30. Audience centric.**

holes" in ARG nomenclatures, or starting points that bring in new audiences from these sources into the rest of the ARG. ARGs are therefore designed to include as many rabbit holes as possible. Sometimes links can point to previous instances and channels e.g. Westworld to allow latecomers to view previous content, but are commonly used to point to new instances and are used to control where audiences go e.g. Dexter.

**Instance patterns.** ARGs often use the Role-play pattern in conjunction with the Live pattern in various ways depending on whether the puppet masters want the audience to contribute to the story, or role-play as pre-existing characters that follow a controlled path. In 19 Reinos, the audience made their own characters and had freedom to name them, choose their family, create their backstory and ultimately decide how they interact with other audience members. We can contrast this with The Black Watchman, where the audience is role-playing as secret operatives that follow a specific path. Role-playing in the former sense is much more common with the Live pattern, as limited rules and rules that can be updated on the fly allow for less restricted rules that appear in static channels, that typically have hard-coded restrictions on how far an audience member can go with their interaction with the fictional world.

The story that is created in experiences that use the Role-play pattern is different for each individual, where the potential story could be the events of the storyworld along with the role-

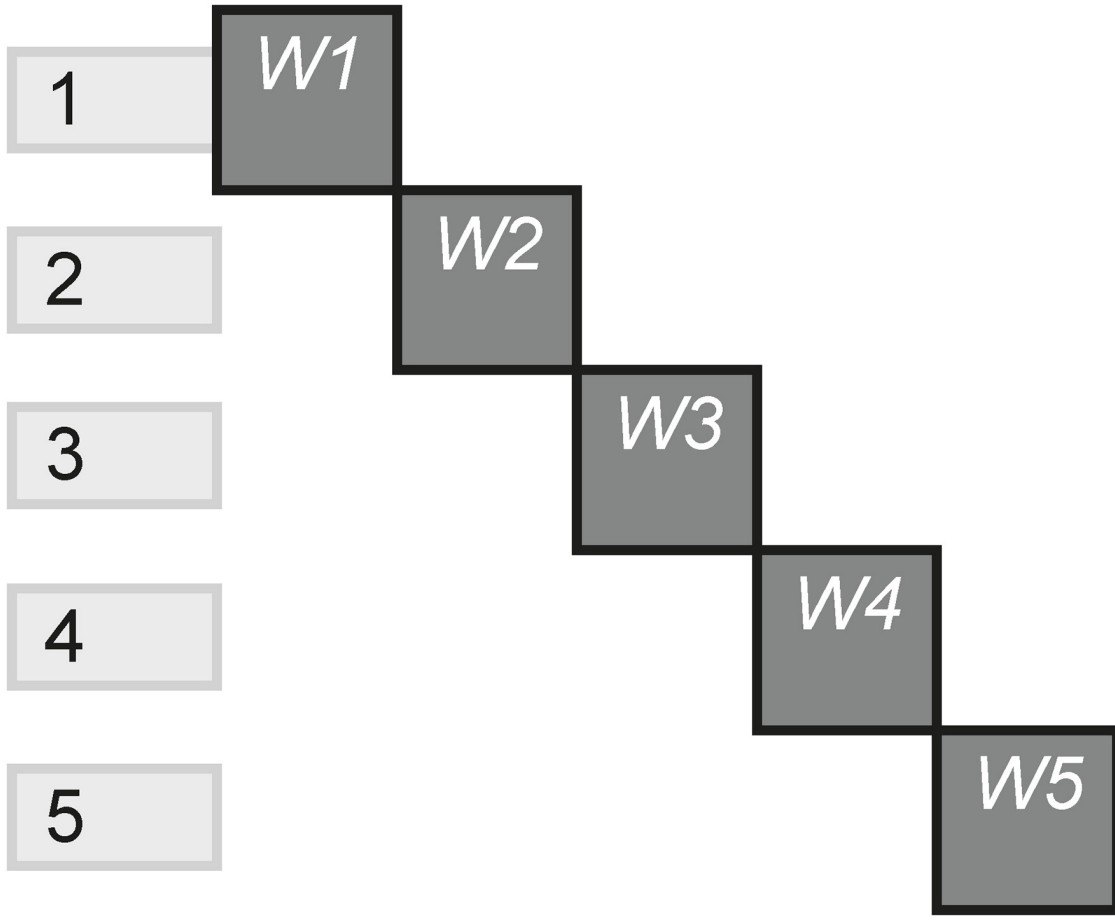

**Fig 31. Live event.**

play behaviour of all characters, or just your own role-play. The issue is that audience member A may never interact or even know audience member B exists, as was the case with many of the audience members of 19 Reinos. Individually, the story they consume is different, but objectively this can be troublesome. An individual looking at the experience from the outside who is trying to find what the story was, will be met with huge amounts of role-play material, some interacting and others not, that may or may not be conflicting with each other. They will have to decide for themselves what the story was, or resign themselves to not include, so far as is possible, the details of individual role-playing content. Similarly, this issue is less troublesome for experiences that limit the role-play to inconsequential or trivial decisions in relation to the story e.g. we know what happens in The Black Watchman because everyone gets the same ending.

## 6.3 Media franchises

**(Harry Potter, The Matrix, Pirates of the Caribbean, Game of Thrones, Pokémon, Overwatch).** A Media Franchise is a collection of media that share a storyworld. Often, there is one primary channel that either sparks the creation or forms the storyworld blueprint of additional media. The primary channel is commonly either a film, television programme, game or book and allows the continuation of characters and events portrayed in these works

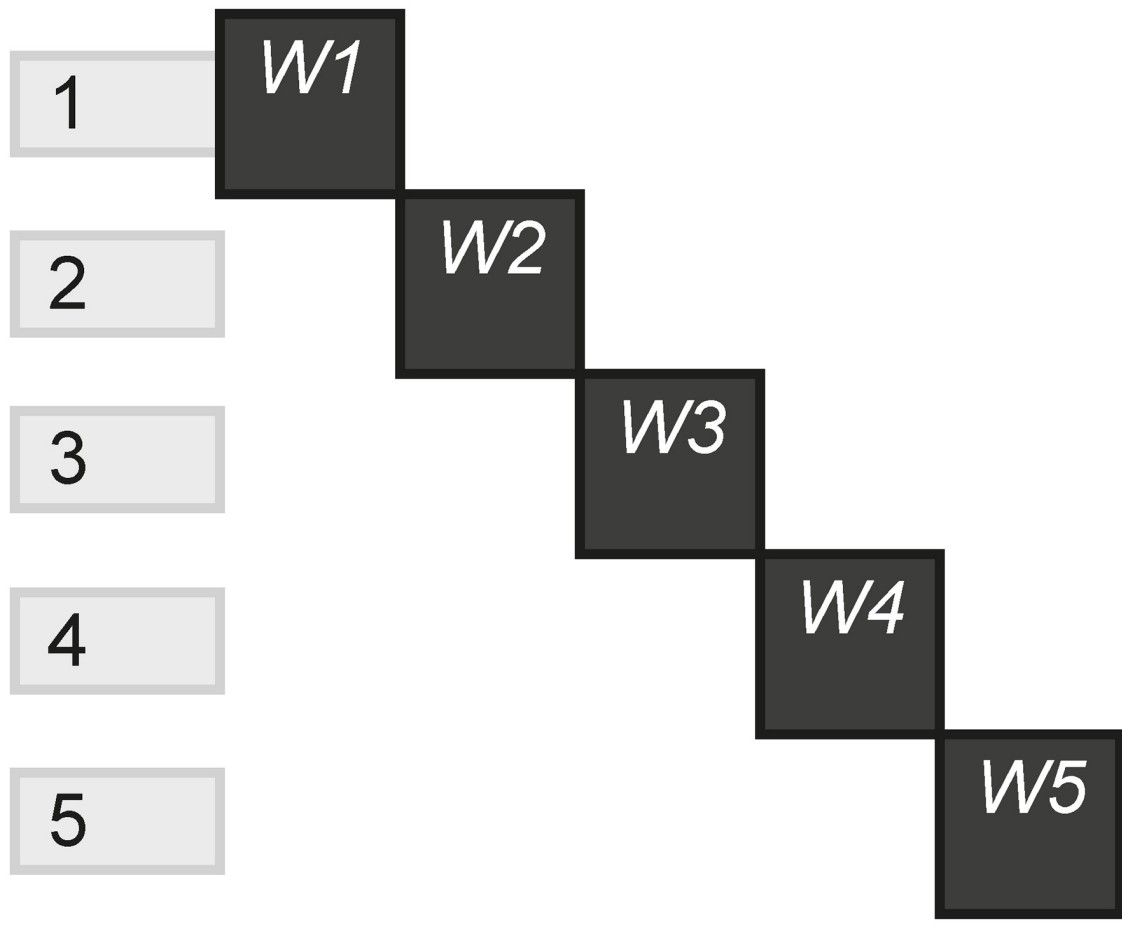

**Fig 32. Artefact.**

in further instalments of the same media, or explores a secondary character or plot thread in a new media.

**Story patterns.** The Many Stories pattern is seen very strongly in franchises, as each channel contains its own story that can be enjoyed without consuming any of the other channels. However, this can arguably be false in some cases, where channels assume that the audience has been previous content. In The Matrix, the third film, although it can be watched and enjoyed to some degree without having consumed any of the previous content, much of the story will not be understood because the audience will not know the importance of The One and the consequences of Neo being identified as such in previous films. Other franchises alleviate this problem by anticipating new audiences by having an introduction in every channel, for example in Pokémon, each channel gives an introduction to the fantasy world and explains what Pokémon are.

**Navigational patterns.** Franchises typically use the Cumulative pattern, releasing new content every few months or years, whilst allowing audiences to consume older content. In some cases, such as Pirates of the Caribbean, multiple channels released at the same time produce a Non-Linear pattern. This occurs when games tied to a film release at the same time, or when spin-off books are published with the release of a film. This sometimes creates a phase effect, in that a franchise will release multiple content in each phase that all share a common

**Table 7. Primary patterns of each form.**

| Form | Primary Patterns | | |
|---|---|---|---|
| | Story | Navigational | Instance |
| Interactive Film/Second Screen | Subsidiary<br>Portmanteau | Non-Linear<br>Linear<br>Connected | Live<br>Artefact<br>Audience-centric |
| Alternate Reality Games | Portmanteau<br>Many Stories | Cumulative<br>Non-Linear<br>Linear<br>Connected | Role-play<br>Live |
| Media Franchises | Many Stories | Cumulative<br>Connected<br>Non-Linear | Role-play<br>Audience-centric<br>Live<br>Artefact |
| Escape Rooms | Portmanteau<br>Subsidiary | Non-Linear<br>Cumulative | Role-play<br>Live<br>Artefact |
| Table Top Role-playing Games | Portmanteau<br>Subsidiary | Cumulative<br>Linear | Role-play<br>Live |
| Exhibits | Portmanteau<br>Many Stories<br>Subsidiary<br>Portmanteau | Non-Linear<br>Artefact | Role-play<br>Live |

time, place, or theme of the storyworld. Phases allow each channel to share marketing momentum at the same time, with each channel promoting the others to varying degrees and reaching the most people by appealing to multiple audiences.

Use of the Connected pattern in a franchise typically indicates intricate plot threads that continue across channels. This is seen in The Matrix Franchise, that has various channels that point to other locations within the franchise. One example is the letter shown in the anime of an incoming attack by the AI on the survivors, that acts as an item that requires acquisition in the game and is received by the characters in the second film. In such cases, franchises start exhibiting characteristics similar to ARGs, with the audience looking out for clues and piecing together the story.

**Instance patterns.** Instance patterns vary in this form from franchise to franchise. There is often a mix of all instance patterns, with the case studies illustrating this. Regarding the Role-play pattern, role-play is usually limited to having a small role in one of the plots in one of the channels. As with other forms, role-play can vary from players creating their own characters that exist in the world e.g. wizards the Pottermore website or trainers in the Pokémon games, to audiences role-playing as pre-existing characters e.g. playing as Captain Jack Sparrow in the games or playing as various characters in Overwatch. Regardless of the role-play involved, this interactivity usually does not have any effect on the other channels. In terms of whether role-play counts as even a small part of canon in the storyworld is perhaps assumed as anything that can happen in the interactive channel, has happened and is part of the canon e.g. you have a limited amount of roleplaying capabilities in Pottermore, that is carefully controlled by software and its underlying rules. However, deeper levels of interactivity can occur if done as the last channel, that does not contradict or conflict with previous channels. In The Matrix Online, players made a major contribution to The Matrix storyworld as characters' role playing in the fictional world set some time after the events of the final film, speaking with main characters and continuing the plot until its eventual finale.

### 6.4 Escape rooms

**(Defenders of the triforce, change the record).**   An escape room, a theatrical entertainment form that has recently become popular, is an experience whereby an audience, usually consisting of small groups, are locked into a room and made to solve clues, progress the story and reach the ending to find the key that lets them out.

**Story patterns.**   Escape rooms predominantly use the Portmanteau pattern as the room, items and actors make up the whole experience, with no one channel taking dominance. However, on a superficial level it can be argued that the Subsidiary pattern is used because all channels rely technologically on the location and room. In some cases, a subsidiary channel such as a hidden CD that relies on there being a CD player is used, but will still be a minor attribute compared to the Portmanteau pattern.

**Navigational patterns.**   Escape rooms make use of all the different types of navigational patterns, with a popular pattern being Non-Linear. In these, players have a choice of which channels they want to consume first, often splitting up the team and assigning individuals or groups to specific channels e.g. Change the Record gave all the channels that contained all the story and clues at the beginning. In other cases, a Cumulative pattern is used to give a more traditional delivery storytelling method by having starting content, middle content and end content e.g. in Defenders of the Triforce, the audience were shown introductory channels, left to complete the puzzles and talk to actors for the bulk of the experience, and then presented with the ending as a performance and a film. A canonical Linear escape room could be one in which each scene is a new room, where the old room is made unavailable to the audience. This gives the producers high control over the experience but assumes that every group of players will progress at the same rate, which can mean the producers stepping in and helping, potentially disrupting the experience negatively.

**Instance patterns.**   Escape rooms commonly make use of the Role-play pattern. In many cases, audiences are expected to role-play as characters in the story, and can theoretically behave as they desire whilst inside the room. However, in terms of story progression, audiences are constrained by what the experience allows you to do e.g. in Change the Record, you are limited to interacting with only the items in the room and in Defenders of the Triforce, actors have scripted responses and will give you the next clue if you converse correctly.

Traditional escape rooms use physical spaces and so use the Live pattern, however digital escape rooms that use virtual reality are becoming increasingly popular. These experiences use the Artefact pattern and in many ways allows the producer deeper control over the audience. Instead of an autonomous body walking around a room in real life, the VR avatar is restricted to the movements, rotations and interactions that the code allows. The producers also lose the risk of interrupting the experience to help the audience, because the time restriction present in real life will not be present whereas the VR application can be replayed and reattempted.

### 6.5 Table top role-playing games

**(DnD).**   Table top games are a type of game that are typically played on a table. They sometimes involve physical objects such as cards, boards and plastic figurines. A sub-set of these are table top role-playing games (table top RPGs), sometimes played with no tangible media channels or online, are games in which a dungeon master (DM) creates quests for a group of players who role-play their characters. Rules are often derived from rule books, which also include information about the storyworld and theme where these quests take place. With the influence of the rule book, the DM may utilise multiple channels to deliver these quests and controls non-playable characters that interact with the players. Stories are different for each group of players, even when quests have been shared amongst DMs,

because the characters' actions, decisions and interactions with characters in the storyworld will be unique for each play through.

**Story patterns.** Table top RPGs mainly use the Portmanteau pattern. With the player character sheet, the document that stores background story, skill attributes and characteristics of a player, the rule book, and the DMs themselves being the channels that make up the experience. Occasionally the Subsidiary pattern emerges when DMs introduce non-essential channels that rely on others such as sound effects, images of characters or documents that purport to be from the storyworld, that enhance the audience's knowledge of the storyworld.

**Navigational patterns.** This form primarily uses the Cumulative pattern. Players have access to their character's sheets, and the rule book at all times, whilst the DM progresses the story with narrative or other Subsidiary channels. However, DM's can also invite the players to choose from a selection which media they want to experience, e.g. asking do you want to watch the video of the dragon or read a description of it? Occasionally, the Linear pattern can be used for times when the DM controls exactly what the players experience e.g. the DM narrates, shows a video, allows characters to be recorded in the characters' sheets, then takes them away and moves on to narrating the starting quest. This is more likely to occur in experiences using the Live Event pattern, where DM can control what the players do more practically.

**Instance patterns.** All table top RPGs use the Role-play pattern, and include role-playing that allows meaningful decisions to be made that affect the characters and the storyworld on a local level. This means that within the confines of the game, the storyworld is permeable, and it is up to the discretion of the DMs, as well as what is legal as stated in the rule book, what is and is not allowed. Everything that occurs inside the game will not have an effect on the storyworld if it is part of a franchise, or other people's games. However, the latter does occur when groups of friends mash their stories and characters together into one coherent world that can be experienced by all the players.

## 6.6 Exhibits

**(Roman baths).** Exhibits can take several different forms including museums, art installations and theme parks. They aim to showcase items, produce a narrative or immerse visitors in a park decorated with a particular theme. They often involve visitors consuming and interacting with multiple channels such as; video projections, live actors, rides, games, posters and items of historical or cultural significance.

**Story patterns.** Story patterns vary depending on the sub-category of the form. In the museums, the story pattern used is usually Portmanteau when artefacts have equal status or Subsidiary when there are primary artefacts. Theme parks tend to be a mix between the Portmanteau and Many Stories pattern, for each ride is a standalone channel, but together make up the storyworld of the theme park. Art installations are usually Portmanteau, and include a variety of channels that are used in a space to make a three dimensional work.

**Navigational patterns.** Exhibits most commonly use the Non-Linear navigational pattern. Visitors are given the choice of which artefact in a museum they want to see e.g. Roman Baths, which direction they want to view the artwork, or which ride to go on first at a theme park. Occasionally, an exhibit will use the Linear pattern to control in what order visitors see different channels e.g. individuals rides at theme parks that may see visitors walking through various themed rooms, then get on a roller coaster, followed by a short performance or film at the end. Exhibits have historically used the Live pattern, but recently there have been experiences that use the Artefact pattern, such as VR theme parks and digital museums.

**Instance patterns.** In exhibits, the Role-play pattern is used differently compared to other forms. Role-play usually involves individuals acting as themselves, but in the context of the

exhibit they are in. At the Roman Baths, live actors roam the museum at certain times, dressed in full Roman regalia, and interact with visitors by giving them details of their Roman life, or details of the Roman Baths. At Disney theme parks, adults pretend that Mickey Mouse and other characters are real whilst some children believe them to be. Every role-play interaction in exhibits is local, and usually has no impact on the experience of other visitors, unless they observe your interactions. Occasionally, the actions of past visitors are recorded in some way e.g. photographs, guest books etc. and can influence the way present visitors enact their role-play.

## 6.7 Comparing applications of patterns

By comparing the way patterns are applied in different forms, we are able to identify situations where techniques from one form can be reapplied to another form. This repurposing can yield various consequences on the form it is applied to, opening up potential novel methods of storytelling.

There are many different ways patterns can be repurposed, such as applying the connected pattern seen in ARGs to franchises, where each new addition to the franchise not only forms part of the story world, but carries on a plot thread or character arc. Others include using the Subsidiary pattern in ARGs the same way as it is used in second screen apps to enhance a primary media channel, using the franchise Cumulative pattern in ARGs, where each experience forms are itself a piece in the larger story world, and using the Linear pattern from interactive films in RPGs to control exactly what the payer sees and is able to interact with.

One such application that will be discussed in more detail is the reapplication of the Role-play and Live Event patterns from ARGs to escape rooms. The Live Event pattern lends itself well to the Role-play pattern in mass audience collective decision making that changes the direction of the story. Examples of this are seen in multiple case study ARGs. In Why So Serious, audiences decided who they wanted to side with and participated in political rallies, observers of the detective's livestream in Dexter decided whether the detective would be victorious or the murderer, and hordes of characters in 19 Reinos battled each other on Twitter to ultimately decide who became king. In escape rooms, collective decision making tends to happen in small parties, where groups may decide to split up and complete their respective chosen channels. The story of the event determines whether this choice is possible, for example Defenders encourages you to work as a team to solve each puzzle and defeat the forces of evil, whereas in other escape rooms it may be efficient to solve the room by assigning people to different channels. However, in Defenders of the Triforce, the room was actually a large hall, filled with several groups of players. Aside from being in the same space, the groups did not interact with each other and the progress and role-play of one group had no impact on any of the other groups. Such escape rooms could use the techniques seen in the above ARGs. In Defenders, the groups could have at one point in the experience worked together to impact how the whole experience ended, instead of having one generic ending. The interaction between the groups could occur in a number of ways e.g. the whole hall working on one puzzle, all groups competing against each other to find a victorious group, decision making by paper like a narrativised ballot or an outcome based on the ratio of groups that completed the puzzles vs those that did not.

## 7.0 Conclusion

In this paper, we have presented the problem associated with the definition of transmedia storytelling and the lack of critical language that can be used to form taxonomies. There is a shortage of tools afforded to us that enables the comparison of different experiences.

We have presented our model and illustrated how it can be used to describe the structural elements of different forms of transmedia storytelling. We then applied our model to twenty case studies and identified three groups of patterns; story, navigational and instance. We then conducted an analyses of the six different forms of transmedia storytelling our case studies fell under, discussing how these forms use the patterns in different ways to achieve certain effects.

These patterns can be used to extend transmedia language and help form taxonomies, by identifying common patterns and their usages amongst various forms of transmedia stories. The main application of this is providing an additional toolset to individuals who wish to 'close read' transmedia stories. This could be done by scholars who wish to deepen their understanding of how different patterns and their usage in various forms change the way particular stories are told. This work may also provide insight into transmedia authors who wish to look back at older experiences and gain inspiration, borrow and modify techniques and usage of patterns.

## Supporting information

**S1 Fig. A transmedia story example.**
(PNG)

**S2 Fig. GoT: 19 Reinos.**
(PNG)

**S3 Fig. APP.**
(PNG)

**S4 Fig. Change the record.**
(PNG)

**S5 Fig. Defenders of the triforce.**
(PNG)

**S6 Fig. Dexter ARG.**
(PNG)

**S7 Fig. TTRPG.**
(PNG)

**S8 Fig. Game of thrones franchise.**
(PNG)

**S9 Fig. Harry potter franchise.**
(PNG)

**S10 Fig. Her story.**
(PNG)

**S11 Fig. Overwatch.**
(PNG)

**S12 Fig. Bandersnatch.**
(PNG)

**S13 Fig. Pokémon.**
(PNG)

**S14 Fig. Pirates of the caribbean.**
(PNG)

**S15 Fig. Prometheus second screen.**
(PNG)

**S16 Fig. Prometheus campaign.**
(PNG)

**S17 Fig. Roman baths.**
(PNG)

**S18 Fig. The black watchmen.**
(PNG)

**S19 Fig. The matrix franchise.**
(PNG)

**S20 Fig. Westworld season 2 campaign.**
(PNG)

**S21 Fig. Why so serious?.**
(PNG)

**S22 Fig. Many stories.**
(PNG)

**S23 Fig. Portmanteau.**
(PNG)

**S24 Fig. Subsidiary.**
(PNG)

**S25 Fig. Linear.**
(PNG)

**S26 Fig. Non linear.**
(PNG)

**S27 Fig. Cumulative.**
(PNG)

**S28 Fig. Connected.**
(PNG)

**S29 Fig. Role-play.**
(PNG)

**S30 Fig. Audience centric.**
(PNG)

**S31 Fig. Live event.**
(PNG)

**S32 Fig. Artefact.**
(PNG)

**S1 Table. Instance state and interactivity.**
(PNG)

**S2 Table. Model of a single channel.**
(PNG)

**S3 Table. Selected case studies.**
(PNG)

**S4 Table. Story patterns summary.**
(PNG)

**S5 Table. Navigational patterns summary.**
(PNG)

**S6 Table. Instance patterns summary.**
(PNG)

**S7 Table. Primary patterns of each form.**
(PNG)

## Author Contributions

**Conceptualization:** Ryan Javanshir.

**Data curation:** Ryan Javanshir.

**Formal analysis:** Ryan Javanshir.

**Funding acquisition:** Ryan Javanshir.

**Investigation:** Ryan Javanshir.

**Methodology:** Ryan Javanshir.

**Project administration:** Ryan Javanshir, Beth Carroll, David Millard.

**Resources:** Ryan Javanshir, Beth Carroll.

**Supervision:** Beth Carroll, David Millard.

**Validation:** Ryan Javanshir.

**Visualization:** Ryan Javanshir.

**Writing – original draft:** Ryan Javanshir.

**Writing – review & editing:** Ryan Javanshir, Beth Carroll, David Millard.

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
