## [Decision Letter · Decision Letter 0]

3 Sep 2019

PONE-D-19-16978

Structural patterns for transmedia storytelling

PLOS ONE

Dear Mr Javanshir,

Thank you for submitting your manuscript to PLOS ONE. After careful consideration, we feel that it has merit but does not fully meet PLOS ONE’s publication criteria as it currently stands. Therefore, we invite you to submit a revised version of the manuscript that addresses the points raised during the review process.

Dear Mr Javanshir,

Many thanks for your submission to the PLOS One 'Science of Stories' collection. As you can see from the attached reviews, a lot of value was found in your work, but also some flaws. On this basis, I am recommending it for potential acceptance pending major revisions. In particular, it seems to me that the following are the most crucial areas for improvement:

1. Your theoretical foundation has a number of significant gaps in it, something all three reviewers noted in different ways, in particular Reviewer 2.

2. The layout and construction of your central argument is based on a perhaps too large sample to be presented as you have chosen to do so.

3. A thorough copy editing is needed.

I hope you will consider revising your article for resubmission.

With every good wish.

Jennifer Edmond, Academic Editor

We would appreciate receiving your revised manuscript by Oct 18 2019 11:59PM. To enhance the reproducibility of your results, we recommend that if applicable you deposit your laboratory protocols in protocols.io, where a protocol can be assigned its own identifier (DOI) such that it can be cited independently in the future. For instructions see: http://journals.plos.org/plosone/s/submission-guidelines#loc-laboratory-protocols

We look forward to receiving your revised manuscript.

Kind regards,

Jennifer Edmond, Ph.D.

Academic Editor

PLOS ONE

2. In order to meet the requirements for the Science of Stories collection, the Guest Editors ask that you please make the code to reproduce your analysis available in a stable, public repository (for example, Zenodo, or GitHub) or a suitable cloud computing service (such as Code Ocean) when submitting your revised manuscript. The code should include a license file and detailed readme so that someone with access to the dataset is able to reproduce your analysis using the code. We ask that you include the DOI for the repository holding your code in an updated Data Availability statement with your revised manuscript.

3. Please expand the acronym “EPSRC” (as indicated in your financial disclosure) so that it states the name of your funders in full.

4. Please ensure that you refer to Figures 3 to 20 in your text as, if accepted, production will need this reference to link the reader to the figures.

5. We note you have included a table to which you do not refer in the text of your manuscript. Please ensure that you refer to Tables 1, 3, 4, 5 and 6 in your text; if accepted, production will need this reference to link the reader to the Tables.

Reviewers' comments:

Reviewer's Responses to Questions

**Comments to the Author**

1. Is the manuscript technically sound, and do the data support the conclusions?

Reviewer #1: Yes

Reviewer #2: Partly

Reviewer #3: Yes

2. Has the statistical analysis been performed appropriately and rigorously? 

Reviewer #1: Yes

Reviewer #2: No

Reviewer #3: N/A

3. Have the authors made all data underlying the findings in their manuscript fully available?

Reviewer #1: Yes

Reviewer #2: Yes

Reviewer #3: Yes

4. Is the manuscript presented in an intelligible fashion and written in standard English?

Reviewer #1: Yes

Reviewer #2: Yes

Reviewer #3: Yes

5. Review Comments to the Author

Reviewer #1: The paper is clear and proposes a useful framework that is demonstrated through 20 examples. There were some in-text citations missing for the example narratives, some misspelled names (e.g., McCluhan), and lack of consistency of capitalisation and italics for the names of titles and the identified patterns. There is also a lack of consistently referring to the “case studies” in the beginning of the paper vs “examples” at the end. Two scholars that have worked on narrative structures and patterns that could be referenced and used as a basis to differentiate this work are Espen Aarseth’s Game Variable Analysis Model and Marie-Laure Ryan’s “Narratives as Virtual Reality 2” book which includes several examples of narrative structures/models.

More explanation of how the authors came up with the terminology for the different patterns (e.g., more stories, portmanteau, and subsidiary) would help contextualise them further and aid other researchers in identifying the same patterns in their own case studies. The authors could add some signposting or introductory paragraphs to contextualise the overall broad patterns that were found and the wider implications of these patterns for future studies.

In the section “6 forms of transmedia storytelling” the authors further explain the narrative patterns in the case studies. It would help solidify the arguments if these sections were integrated with each case study’s graphical pattern rather than as its own separate section at the end so that the reader does not have to scroll back and forth and the rationale and identified pattern would be clearly reinforced.

Reviewer #2: Overall I admire and appreciate what the authors are trying to do here. They are right that the term transmedia is broad and takes on different meanings in different concepts. More consistent language could therefore the valuable. The article is well written and clearly structured, with some useful terms.

However, I cannot help but feel that the authors are overlooking a vast amount of recent research into transmedia studies that does some of this work for them. The authors rely on understandings of transmedia from 2006-2009, but ignores almost everything else. Transmedia studies has done much work since then, employing all kinds of new terms, models and approaches. For example, look at:

- Susana Tosca and Lisbeth Klastrup's work

- Matthew Freeman's work

- Routledge Companion to Transmedia Studies

- Elizabeth Evans' work

- Renira Rampazzo Gambarato's work

All of these, and others, have developed new models and terms re transmedia, and these should be acknowledged if the author's aims are genuine and to be of significance. Without engaging with this work, the the article feels outdated - though admirable in its intent.

The other issue, for me, is the number of case studies. 20 is surely too many for qualitative analysis, which leaves very little room for detailed analysis. Perhaps more of a quantitative approach would be better? In fact, very little research in transmedia studies takes a quantitative approach, so this would make it more original.

Reviewer #3: This article deals with an original analysis of Transmedia productions and strategies, offering new methodologies and a pattern which could easily be reproduced.

The methodology is well described and we can easily follow the arguments of the authors throughout the article.

The authors used different cas studies, allowing them to draw some conclusions and provide a relevant pattern.

We would have liked to have a final table at the end of the article, presenting a summary of the projects and findings. It could help the reading.

The authors should give their own definition of Transmedia storytelling and Transmedia projects, since the case studies they have chosen to use in this article do not match Jenkins' seminal definition.

There are still some mistakes and misspellings in the text.

6. PLOS authors have the option to publish the peer review history of their article (what does this mean?). If published, this will include your full peer review and any attached files.

Reviewer #1: Yes: Nicole Basaraba

Reviewer #2: No

Reviewer #3: Yes: Melanie Bourdaa

---

## [Author Response · Author response to Decision Letter 0]

10 Nov 2019

1. Grammar - We have edited the document to fix grammar and spelling mistakes. (Reviewer 1, Reviewer 2, Reviewer 3)

2. References and Background – We have updated our background section to include more context to our research, and further related work. (Reviewer 1)

3. Signposting – We have expanded our introductory paragraph in section 5 to provide more signposting for the reader. (Reviewer 1)

4. Integrate section 6 with case studies – We felt that this would in fact make the reading much harder, and would risk talking about the same things twice. Our findings in section 6 are grouped according to the forms of the case studies. (Reviewer 1)

5. Table – We have added a table in section 6 to summarise our findings (Reviewer 3).

6. Definitions – We have included more background literature for definitions, and included our own (Reviewer 3).

7. Case study amount – We feel that we needed at least this amount to be able to identify the patterns that we did. Our objective was not to do a detailed qualitative analysis, but rather a structural analysis of how these experiences operate. (Reviewer 2)

---

## [Editor Report · Decision Letter 1]

15 Nov 2019

Structural Patterns for Transmedia Storytelling

PONE-D-19-16978R1

Dear Dr. Javanshir,

We are pleased to inform you that your manuscript has been judged scientifically suitable for publication and will be formally accepted for publication once it complies with all outstanding technical requirements.

With kind regards,

Jennifer Edmond, Ph.D.

Academic Editor

PLOS ONE

Additional Editor Comments (optional):

Dear Mr Javanshir,

Thank you for your revised manuscript, and for the attention you have given to the reviewer feedback. I am happy to recommend acceptance on this basis.

With every good wish,

Jennifer Edmond
---

## [Editor Report · Acceptance letter]

25 Nov 2019

PONE-D-19-16978R1 

Structural Patterns for Transmedia Storytelling 

Dear Dr. Javanshir:

I am pleased to inform you that your manuscript has been deemed suitable for publication in PLOS ONE. Congratulations! Your manuscript is now with our production department. 

With kind regards,

on behalf of

Dr. Jennifer Edmond 

Academic Editor

PLOS ONE